# CHIRP: A Fine-Grained Benchmark for Open-Ended Response Evaluation in Vision-Language Models

## Abstract

The proliferation of Vision-Language Models (VLMs) in the past several years calls for rigorous and comprehensive evaluation methods and benchmarks. This work analyzes existing VLM evaluation techniques, including automated metrics, AI-based assessments, and human evaluations across diverse tasks. We test our model on a scaling suite across vision and langauge sizes, identifying shortcomings of current evaluation approaches. To overcome the identified limitations, we introduce *CHIRP* - a new long form response benchmark we developed for more robust and complete VLM evaluation. We provide open access to our evaluation code and the CHIRP benchmark to promote reproducibility and advance VLM research.

## 1 Introduction

Recently, a lot of significant advances have been made in Vision-Language Models (VLMs), driven by breakthroughs in computer vision and natural language processing Chen et al. (2022); Li et al. (2023b); Liu et al. (2023b); Sun et al. (2023). However, existing VLM benchmarks, often designed for specific tasks (e.g., VQAv2 Goyal et al. (2017)), struggle to accurately reflect real-world VLM performance and capture nuanced differences between models Hsieh et al. (2024). This is particularly evident when evaluating models with significant architectural variations, where standard benchmark scores remain similar despite noticeable differences in human-perceived model quality.

To address this issue, we introduce CHIRP, a hybrid VLM benchmark that combines automated metrics' scalability with human evaluators' nuanced judgment. We argue that this approach is crucial for capturing the complexities of VLM behavior, which traditional benchmarks often fail to represent.

To demonstrate the limitations of existing benchmarks and the efficacy of our proposed method, we use a scaling suite called Robin, a series of VLMs trained at various scales, inspired by the Pythia language model suite Biderman et al. (2023). By systematically varying the Vision Encoder (VE) and the Large Language Model (LLM) sizes, we see that while some benchmark scores remain largely unaffected, human evaluations reveal significant differences in the models' outputs quality.

Our findings underscore the need for more robust and human-centric VLM evaluation methodologies. CHIRP paves the way for developing more reliable and informative VLM benchmarks, ultimately leading to the creation of more effective and impactful VLMs. Furthermore, we will also investigate in detail the possibility of using different AI agents as proxies to human evaluations.

## 2 Related Work

**Scaling Suites.** Scaling laws have recently emerged as one of the central research areas in large foundation models Aghajanyan et al. (2023); Isik et al. (2024). These laws enable performance prediction based on variations in compute time, dataset size, and model parameters, facilitating efficient resource allocation by extrapolating results from small-scale experiments.

Kaplan et al. (2020) pioneered the application of scaling laws to language models, demonstrating a power-law relationships between loss and model size, dataset size, and compute time. This has led to practical applications, such as the Pythia suite Biderman et al. (2023), which comprises of identically trained language models with varying parameter sizes, empirically verifying these scaling laws.

Cherti et al. (2023) investigated the scaling laws of the CLIP vision encoders, training and comparing different sizes of the CLIP vision encoders on the same data. These models indeed verified the aforementioned scaling laws and have become a very popular suite of models.

**AI-based Evaluation.** The advent of powerful foundation models like GPT-4V offers a new way to evaluate weaker models, moving beyond traditional, rigid metrics such as exact string matching, as done in popular benchmarks like Hudson & Manning (2019); Mishra et al. (2019); Singh et al. (2019). Early evidence from benchmarks like MM-Vet Yu et al. (2023) and VQA tasks Agrawal et al. (2016) suggests that evaluating with stronger models offers a promising path towards more comprehensive and insightful evaluation, surpassing the limitations of static, string-based methods Ji et al. (2023); Lee et al. (2024). This shift towards leveraging the semantic understanding of LLMs for evaluations promises to unlock a better evaluations, leading to a better understanding of model capabilities.

Zheng et al. (2023) introduce two benchmarks, MT-Bench and Chatbot Arena, to explore the feasibility of employing LLMs as judges. Their findings indicate that advanced LLMs, such as GPT-4, closely align with human preferences, achieving over 80% of agreement Rafailov et al. (2024). Similarly, AlpacaEval Li et al. (2023a) utilizes LLMs to assess instruction-following models.

Wu & Aji (2023) focused on the bias in evaluations conducted by both human and LLM annotators, particularly noting a preference for flawed content if it avoids brevity or grammatical errors, and introduced the Multi-Elo Rating System (MERS) for more nuanced assessments. A study by Koo et al. (2023) pointed out significant biases of LLMs evaluators, with an average Rank-Biased Overlap (RBO) score of 49.6%, suggesting a misalignment between machine and human preferences.

While there exists benchmarks like LLaVA-Bench Liu et al. (2023a) which elicit open-ended responses, and evaluate using LLMs, they often lack thought-provoking and diverse questions that challenge a model's perception and contextual understanding. For example, most of the open-ended questions in LLaVA-Bench are of the form: "describe the image...". We address this shortcoming with our benchmark CHIRP.

## 3 CHIRP

We introduce CHIRP, a new evaluation benchmark, which grades long form responses. CHIRP comprises of 104 open ended questions, evaluated by either humans or VLMs. These free form questions do not correspond to a single "correct" answer. Instead, they require models to generate flexible, creative and complex responses. Consequently, we evaluate models using a preference based rating in which two model's responses are compared side by side. Instructions on downloading the CHIRP benchmark can be found in Appendix C.

We will show in Section 5.2 that current benchmarks evaluate models with high variance, making it difficult to accurately rank models with similar performance. CHIRP addresses this issue by using significantly fewer but higher-quality prompts and pairwise evaluations to establish a more reliable ranking.

### 3.1 Generating the dataset

We wrote questions along with image descriptions, which were then refined with the help of GPT-4 OpenAI (2023). The image descriptions were given to Dalle-E 3 to generate the associated images. We finetuned the image descriptions and questions by hand to generate the optimal image question pairs.

The questions created are classified in 8 distinct categories: **descriptive analysis, inferential reasoning, contextual understanding, emotional and psychological understanding, ethical evaluations, abstract understanding, creative and subjective analysis, and visual aesthetics evaluation.** Detailed descriptions and examples of these categories can be found in Appendix C.1.

Unlike many datasets that rely on pre-existing images, our approach allows us to generate images specifically tailored to our thought-provoking questions and detailed analysis. This also removes the risk of the model having seen the image in training. Moreover, we eliminate the risk of evaluating models on contaminated images as all of them were validated by hand. A sample text-image pair of the CHIRP dataset can be seen in Appendix C.2.1.

### 3.2 Human based evaluations

We utilize CloudResearch for large scale human evaluation of our model's responses. To this end, we presente users with the responses of two models to the same question and ask them to indicate their preferred response on a set of criteria. There are 5 criteria: **overall preference, relevance and completeness, understanding and reasoning, hallucinations, and details**. These criteria were chosen as empirical evidence showed that these were the most important to a user's perception of the model quality. An example of the user interface as well as a detailed description of each criterion can be found in Appendix C.2.1.

### 3.3 VLM based evaluations

To evaluate our models on CHIRP at scale, we experiment with the use of VLMs, namely GPT-4V and LLaVA-34B, as judges. Rather than asking a human for model preferences, we asked the VLMs to indicate their preferred response for each criterion. For GPT-4V, we utilized two distinct prompts: **GPT-4V (S)** (simple), which directly solicited model preferences, and **GPT-4V (R)** (reasoning), which prompted the VLM to reason before making a decision. We extracted the VLMs final choice using GPT-3.5. Detailed explanations of these prompts are provided in Appendix C.2.3.

### 3.4 Elo ratings

To benchmark our models using CHIRP, we calculated Elo scores based on the evaluators' indicated preferences. Because Elo calculations are not order-agnostic, we performed 500 bootstrap iterations for each Elo score.

## 4 Experimental Setup

To evaluate CHIRP's effectiveness, we test it on a suite of VLMs with controlled single-parameter variations. This setup allows us to assess whether CHIRP can capture subtle differences in model performance that existing benchmarks fail to detect. We outline the details of the *models evaluated on* and the *benchmarks compared against* below.

**Evaluation models**   We test CHIRP on a suite of models based on the LLaVA architecture, with controlled variations in both the LLM size and the Vision Encoder (VE) size. Each model is built by combining an LLM from the Pythia suite with a VE from the CLIP suite. These models, referred to as Robin, include three main experimental groups:

1. *All Robin* models - 20 models with 5 Pythia sizes (410M, 1.4B, 2.8B, 6.9B, and 12B parameters) paired with 4 CLIP models (Base, Large, Huge, and gigantic)

2. *LLM Size* ablation - ablate the Pythia model size across the Robin models with the gigantic CLIP vision encoder (ViT-g)

3. *VE Size* ablation - ablate the CLIP model size across the Robin models with a 12B parameter Pythia LLM

By running experiments across these Robin models, we aim to demonstrate CHIRP's ability to capture subtle differences in model performances that are often missed by traditional benchmarks but have noticeable differences in response quality. We illustrate this complete pipeline in Figure 1.

More on the architecture and training procedure for these models can be found in Appendix A.

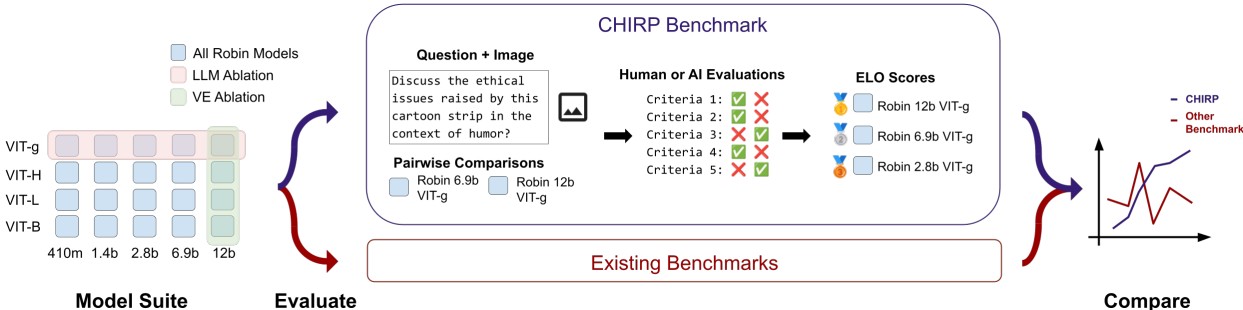

Figure 1: **Experimental Setup Pipeline.** Detailed pipeline used to evaluate CHIRP: from the left we have the models trained with the 3 different ablations, then in the middle we evaluate said models on different benchmarks, including CHIRP, and finally we compare the model rankings obtained on the different benchmarks.

**Benchmark Comparisons**  We ran our suite of models on the following benchmarks: ScienceQA Lu et al. (2022), GQA Hudson & Manning (2019), VQAv2 Goyal et al. (2017), TextVQA Singh et al. (2019), MM-Vet Yu et al. (2023), and LLaVA-Bench Liu et al. (2023b).

We calculate a scaled average to show the general results from all benchmarks. The scaled score is calculated as follows: let $S$ be the matrix of scores, with each row $S_{i,:}$ representing the scores model $i$ obtained on all $N$ benchmarks, and $S_{:,j}$ representing the scores of all models on benchmark $j$. Let $S^*$ be the scaled scores vector.

$$S_i^* = \frac{1}{N} \sum_j \frac{S_{i,j} - min(S_{:,j})}{max(S_{:,j}) - min(S_{:,j})}$$

**Running Robin on CHIRP**  For human based evaluations, we validate this method by evaluating our suite of models on all five CHIRP criteria across the *LLM size* and *VE size* ablations. Due to limitations in time and budget, for each question of the dataset, we randomly sample five model matchups out of all the possible model pairwise combinations. We also ran evaluations across our entire suite of VLMs to judge the overall preference criteria. To this end, we randomly selected 25 matchups from the 190 possible pairs of Robin models. Full details on the human evaluation setup can be found in Appendix section C.2.2.

For AI based evaluations: we evaluate all combinations of matchups from the *LLM size* and *VE size* ablations across all criteria. We also run **GPT-4V (R)** evaluations on a random sample of 50 matchups from all combinations of Robin models on the overall preference criteria.

## 5 Evaluation Results

### 5.1 Comparison with existing benchmarks

In all benchmarks, there is no discernible relationships between VE size and model performance. The scaled average of the results are plotted in Figure 2. As it is shown, there is no clear relationship between VE size and model performance. However, a slight trend between LLM size and performance is observed. The complete results of the models on these benchmarks are detailed in Appendix A.4, which includes a complete score table and heatmaps for all benchmarks. In comparison, a preliminary analysis of the results on CHIRP clearly show the effect of language model scaling.

### 5.2 Investigating differences between CHIRP and other benchmarks

Empirically testing our models we saw that model performance aligned more closely to the trends indicated using CHIRPs evaluations. Here we aim to determine the source of why existing benchmarks might not

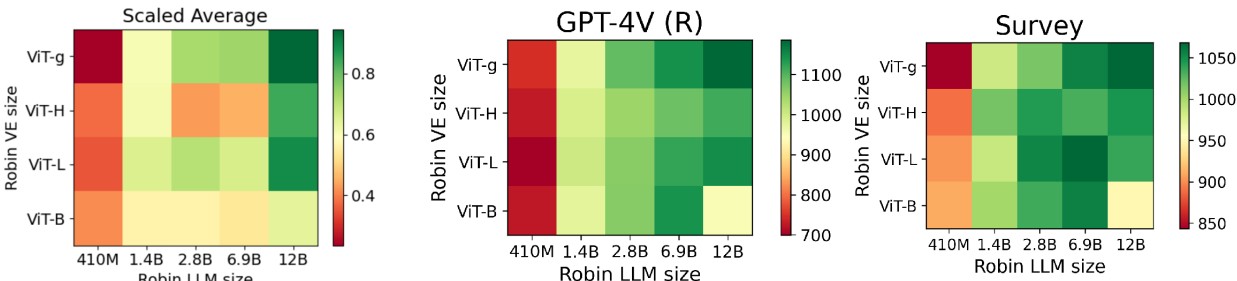

Figure 2: *All Robin* model performance on existing benchmarks vs CHIRP. **Left.** Heatmap showing the scaled average score detailed in Section 4. **Center.** Elo calculated from GPT-4V (R) on CHIRP. **Right.** Elo calculated from human survey on CHIRP.

capture all observed model capabilities, and in doing so highlight the gap in benchmarking that CHIRP hopes to fill.

We hypothesized that existing benchmarks may not capture subtle changes between models for one of the following reasons:

1. short responses don't convey enough information to thoroughly evaluate model performance
2. benchmarks aren't graded with sufficient accuracy
3. questions don't demand a detailed examination of images or their ground truths may not be accurate

In the following subsections, we test each of the above hypothesis to see if addressing these issues yields results more closely aligned with those of CHIRP and if it could help improve the targeted benchmarks. To do so, we sample 100 questions from both the GQA and textVQA benchmarks.

### 5.2.1 Long vs Short Responses (LvSR)

Most benchmarks were evaluated on short responses; with explicit instructions to "respond with one word or phrase". However, we hypothesize that short responses do not convey sufficient information to evaluate model performance in detail. To test this theory, we allowed models to generate longer responses without prompting for brevity. We then collected, manually evaluated, and compared these LvSR to see if they offered a more nuanced assessment of the models.

The GQA benchmark provides an evaluation script that grades responses using string matching on single phrase responses. On the sample of 100 GQA questions, we prompted and manually graded our models for LvSR to see if new trends across the *LLM size* ablation appear with longer responses, the results of which are shown in Figure 3. For sufficiently large models, we did not notice a significant improvement in overall model accuracy. However, models often got different questions correct when responding with LvSR. To show this, we calculated a superscore, in which responses were marked correct if either the long or short response was correct (See Figure 3). The improved results of the superscore indicate that while long and short responses achieve a similar overall accuracy, they tend to be accurate for a different set of questions. This suggests that evaluating long and short responses demonstrate different model skills. In Appendix B.4.3, we've included examples of differing responses when prompting for long and short answers. We believe that this is one of the reasons CHIRP indicates different evaluation results as compared to existing benchmarks.

### 5.2.2 Inaccurate Grading

**LLM Grading.** Most existing automatic metrics are incapable of evaluating longer responses, and often fail in scenarios where the models being tested do not output the answer in the expected format, as shown in Hudson & Manning (2019); Singh et al. (2019). For example, models may respond with a synonym for the ground truth, which can cause issues with exact string matching based evaluations. These issues can be especially prevalent with non instruction tuned models, or small scale models.

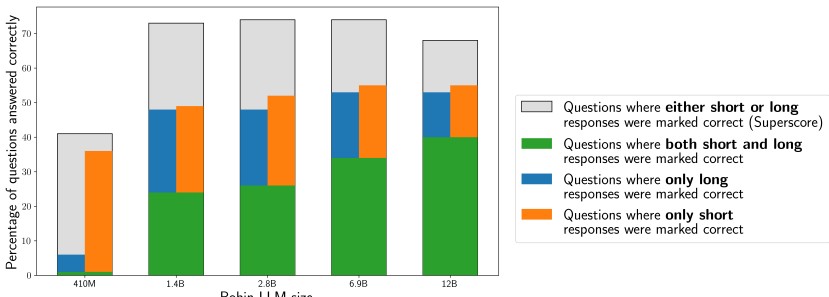

Figure 3: Accuracy of long vs short responses on GQA sample for *LLM Size* ablation.

To address responses that automated evaluations cannot recognize, we utilize GPT-4 OpenAI (2023) to evaluate whether a given response matches the correct answer or not. We run this LLM evaluation on both the long and short responses, using the prompting detailed in Appendix B.3.

On short responses, LLMs tend to mark more answers as correct when compared to existing automated evaluations. An example of this behaviour can be found in Appendix B.4.3. By comparing LLM evaluations to manual evaluations of LvSR in Figure 4, we calculated the accuracy of LLM evaluations on *LLM size*. This analysis shows that LLM evaluations can be slightly more accurate than automated evaluations, though not enough to reveal new model capabilities.

**VLM Grading.** Our empirical analysis revealed that ground truth answers are not always representative of all possible correct answers. In GQA and TextVQA, this issue arises from ambiguous questions that can have multiple valid answers, as shown in Appendix B.4.1 and B.4.2. In questions where ground truth answers don't encompass all valid answers, LLMs don't have sufficient information to accurately respond.

We explore using stronger VLMs, namely LLaVA-34B Liu et al. (2023b) and GPT-4V OpenAI (2023), to evaluate our models responses in order to account for such cases. We ask the VLM to individually evaluate each model's long response, question by question. The exact prompts used for LLaVA-34B and GPT-4V are in Appendix B.3.

**Human Grading.** We manually grade the responses generated from models in the *LLM Size* ablation. We use these grades to calculate the accuracy of the automated, LLM and VLM evaluations. The accuracy is determined by the percentage of responses where the evaluation agrees with the human grading. The accuracy is plotted in the center plots of Figure 4.

**Comparison of Evaluation Methods.** We graph scaling across *LLM size*, and *VE size* using the automated and AI evaluation methods in Figure 4.

We observe that long-form response accuracy exhibits greater sensitivity to model parameter count. Furthermore, prompting short-form benchmarks for extended responses does not enhance grading accuracy. These findings highlight the necessity of explicitly designing benchmarks for long-form prompting, a challenge we address with CHIRP.

Additionally, our findings indicate that neither AI-based nor traditional automated evaluations show a consistently stronger agreement with human assessments, as evaluation accuracy remains inconsistent across short and long responses, with no method proving definitively superior. This suggests that CHIRP's improvements stem primarily from its design favoring open-ended questions over those with ground-truths.

### 5.2.3 Poor questions and ground truths

To rigorously assess the reliability of existing benchmarks, we sampled 100 random questions from GQA as well as 100 from TextVQA. These questions require the model to observe the image and answer objective

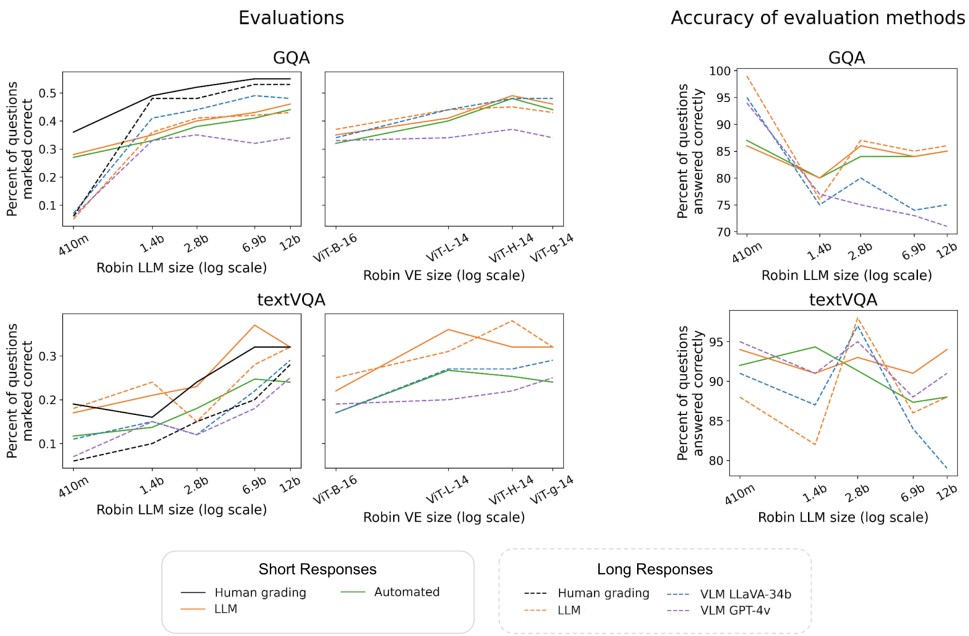

Figure 4: **Left.** Evaluating GQA and textVQA across the *LLM size* and *VE size* ablations using automated string matching programs, LLMs, VLMs, and humans evaluators. Solid lines are evaluations of responses where models were prompted for short responses. Dashed lines have no such instruction. **Right.** Accuracy of these evaluation methods on the GQA and textVQA sample over the LLM size ablation. The accuracy was determined by comparing evaluations to the human grading.

facts. We examined the questions, the provided ground truth answers, and model responses across all model size combinations.

In 100 questions sampled from GQA, we found that 9 questions had incorrect ground truth answers. If we want to estimate the error of this value, the actual percentage of incorrect prompts $\hat{p}$ is $\hat{p} \in p \pm z * \sqrt{\frac{p*(1-p)}{n}}$. For a 95% confidence interval: $z = 1.96$, and with our sample size $n$ of 100, we measured $p = 0.09$, we are 95% certain: $3.4\% \leq \hat{p} \leq 14.6\%$. This equates to 770,769 to 3,309,773 questions of the 22,669,678 GQA questions being incorrect. Although this is a large spread, this result remains quite significant, as most improvements on State of The Art (SoTA) models are very small, regularly under 3% Li et al. (2023b). These findings lead to the conclusion that if 2 models score within 3% of each other on GQA, they could very well be equal in actual performance on it. Representative examples of the aforementioned questions are shown in Appendix B.4.1.

Conducting the same study for TextVQA, we identified only 5 problematic questions in the sample that either did not require reading the text in the image, or were too vague and did not correspond to a clear correct answer. Redoing our previous calculations, we conclude with 95% certainty that $0.73\% \leq \hat{p} \leq 9.27\%$. Although SoTA models are indeed close in performance, we are not as confident as in the case of GQA. However, two SoTA models scoring within 0.7% of each other on TextVQA can be considered equally good on the benchmark. Representative examples of the aforementioned questions are shown in Appendix B.4.2.

This leads us to our main conclusion for this section. Errors in ground truths lead to inaccurate benchmark scores. Additionally, the fact that most modern SoTA models fall within the margin of error of these benchmarks highlights the need for a precise, pairwise, benchmark like CHIRP.

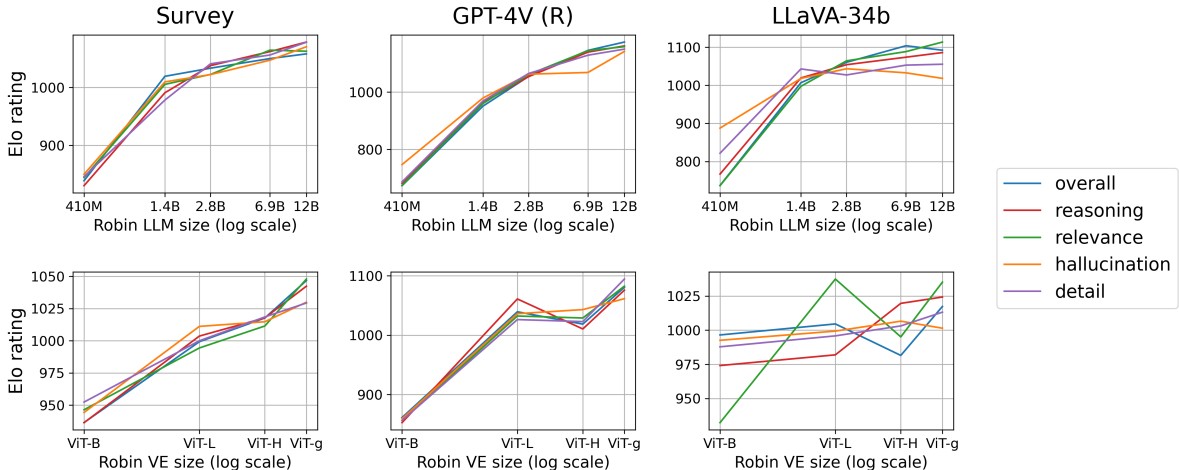

Figure 5: Mean Elo calculated over LLM Size (top row) and VE Size (bottom row) using different evaluators (columns) and criteria (series). Graphs are calculated using bootstrapping on 1000 samples.

# 6 Using CHIRP in practice

We have not yet discussed the practicality of using our benchmark. In general, human evaluations are expensive, and thus are not the standard method for evaluating model performance. Here we investigate the extent to which AI evaluations can be used as a proxy for human evaluations in CHIRP.

## 6.1 Qualitative Analysis - Trends

The results from the human and VLM evaluations using average Elo rating is shown in Figure 5.

With regards to the *LLM size* ablation, we note a clear scaling trend, with all the evaluators ranking the bigger models the best performers across all categories. However, we do note that the biggest marginal improvement occurs from the 410M Pythia-based Robin to the 1.4B Pythia-based Robin.

With regards to the *VE size* ablation, only the human survey results exhibit a strictly monotonically increasing trend with scale. Indeed, AI evaluations of CHIRP do not correlate VE size with model performance. GPT-4V (R) evaluations of CHIRP demonstrate some scaling with model size, with ViT-L performing surprisingly well. To the contrary, LLaVA-34B gives a very consistent score to all models across all categories, with the exception of the "hallucination" evaluation where the trend is similar to the one from GPT-4V (R). It is worth noting however that human surveys exhibit high variance in Elo trends, mostly due to different evaluators having very different preferences.

The heatmap of median Elo scores in Figure 2 allows us to directly compare GPT-4V (R) and human surveys results. In the following sections, we will explore why GPT-4V (R) evaluations seem to capture some trends more distinctly while not others.

## 6.2 Quantitative Analysis - Agreement

To evaluate the efficacy of AI evaluations, we first examine the agreement between AI and human preferences. To this end, we use Cohen's Kappa.

### 6.2.1 Cohen's Kappa

Cohen's Kappa Cohen (1960) is a method used for calculating inter-rater reliability, that takes into account random chance agreement. A Cohen's Kappa score of 1 indicates a complete agreement between reviewers, while a Kappa of 0 indicates no agreements other than a random chance of agreement. Further details on the

calculation of Cohen's Kappa can be found in Appendix C.2.5. Looking at Table 1, the results indicate that both GPT-4V (S) and GPT-4V (R) have higher agreement with human surveys compared to LLaVA-34B. We also note that GPT-4V (R) exhibits the most agreement to the human surveys of the both of them. However, according to Landis & Koch's interpretation of Cohen's Kappa Landis & Koch (1977), GPT-4V (R) only achieves "slight" to "fair" agreement. Despite the low overall agreement, GPT-4V evaluations still exhibit very similar trends to human evaluations.

Table 1: Agreement and Cohen's Kappa between human surveys and AI evaluations across 2 studies

|  | *LLM size* ablation | | *VE size* ablation | |
| --- | --- | --- | --- | --- |
| Models | Agreement | Cohen's Kappa | Agreement | Cohen's Kappa |
| GPT-4V (S) vs human | 67.5% | .10 | 63.7% | .204 |
| GPT-4V (R) vs human | 69.3% | .114 | 64.5% | .216 |
| LLaVA-34B vs human | 60.8% | .014 | 50% | 0.0 |

### 6.2.2 Model Size Agreement

For each of our evaluation methods, we calculated the frequency with which the evaluator preferred the larger model in any given matchup. The results in Table 2 show that GPT-4V evaluations favor models with more parameters more frequently than human evaluators. We also see that although users tend to prefer larger models, this is not as systematic as we had initially believed.

Table 2: Model size agreement by method

| Method | *LLM size* | *VE size* |
| --- | --- | --- |
| Human Survey | 68.5% | 61.3% |
| LLaVA-34B | 66.5% | 53.0% |
| GPT-4V (S) | 76.5% | 65.2% |
| GPT-4V (R) | 79.4% | 66.4% |

### 6.2.3 Contradictions

One hypothesis for why trends are better captured using AI evaluations is that a single AI evaluator is more consistent than the combined evaluations of many different humans, as different humans may have different preferences or leniency. We tried negating this by aggregating multiple human surveys together however it is possible this still influenced the results. To evaluate the consistency of AI versus human evaluators, we introduce a concept to measure contradictions in their rankings. A contradiction occurs when an evaluator's preferences form a cycle, such as preferring A over B, B over C, but then C over A. A more exhaustive explanation along which sample graphs is given in Appendix C.2.8. This inconsistency suggests a lack of transitivity in their judgments. By counting these contradictions, we can determine how reliably an evaluator ranks models.

The results presented in Figure 6, indicate that human and LLaVA-34B based evaluations tend to have the most contradictions, requiring more runs to average out human or model inconsistencies. GPT-4V (R) however is the model with the least contradictions, leading us to the conclusion that a single run is sufficient as the model is highly consistent in its responses.

### 6.3 Observations and Insights

Although AI-based evaluations don't consistently agree with human evaluations on a case-by-case basis, GPT-4V (R) displays both higher agreement with humans preferences and less contradictions than GPT-4V (S).

In general, GPT-based evaluations tend to produce lower variance results which correlate better with training loss, as shown Appendix C.2.6. We hypothesize that these smoother results are attributed to the fact that GPT employs a more consistent approach to grading across evaluations, whereas multiple different human evaluators lead to more variability, as indicated by the higher rates of contradictions. This variability could

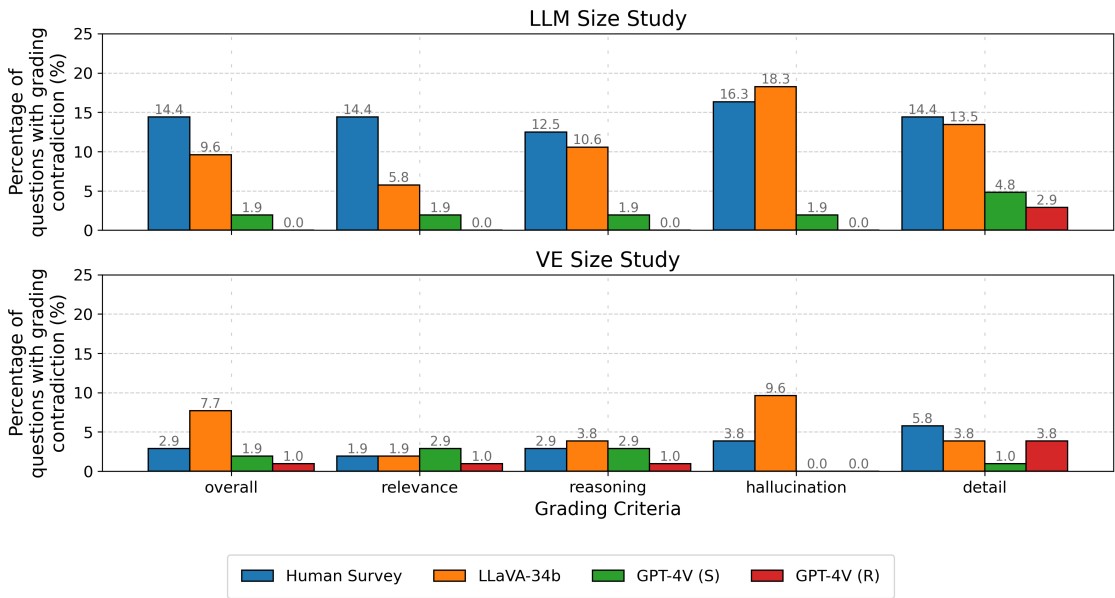

Figure 6: Percentage of CHIRP questions graded with a contradiction of preferences within a specific criteria. **Left:** contradictions found in the LLM ablation study. **Right:** contradictions found in the VE ablation study.

be affecting our ability to accurately measure how well human evaluations correlate with training loss and we hope to address this in future work.

Another possibility is that AI evaluations favor models with larger LLMs because the LLMs generate preferable strings of words irrespective of the content of the image. However, we rule out this possibility by showing that GPT-4V (R) preferences do not align with the more likely logit probabilities of question-answer strings in Appendix C.2.7.

Furthermore, there seems to be an ideal ratio of VE size to LLM size that provides an optimal model, which will be the preferred model for that LLM size. Although this relationship was hinted at in both the loss and previous benchmarks, it was made more apparent in the human preferences result in CHIRP. As illustrated in the complete graph of human preferences shown in Appendix C.2.9, models with larger VEs perform poorly when paired with the smallest LLM, and similarly, larger LLMs struggle when paired with the smallest VE. This trend, which was not evident in other benchmarks, is now made clear through CHIRP's evaluations.

Ultimately, both human and AI evaluations show that performance on CHIRP correlates with loss more than other evaluation tasks. We take this as evidence that the CHIRP benchmark assesses a valuable and unique skill that other benchmarks do not test for. This makes CHIRP a useful addition to the suite of benchmarks that is currently used to evaluate VLMs.

## 7 Limitations

Although the CHIRP benchmark revealed scaling trends in our models that other benchmarks did not, it has several notable limitations. First, it heavily relies on the strong language proficiency of the evaluator, to evaluate a models' perceptual capabilities. This is particularly relevant to frontier AI models, which will require human evaluations.

Second, the benchmark is not very extensive as it only contains 104 questions on 104 images. However, the small size is a deliberate choice based on the cost of evaluations. As grading the responses requires VLM or human evaluations, cost is a major consideration when deciding the size and 104 was seen as a good balance between evaluating the models performance and the cost or evaluating. This is in line with other small,

high quality, and well respected datasets like MM-Vet Yu et al. (2023), 218 questions on 200 images, and LLaVA-Bench Liu et al. (2023b), 60 questions on 24 images, which both require LLM evaluations, which itself is cheaper than VLM or human evaluations.

Finally, models are benchmarked via pairwise matchups. Therefore models can only be compared via a direct matchup or mutual matchups. This requires more work when validating a new model, requiring matchups which each of the most performant models, however we believe this is a valuable trade-off for a considerably more accurate evaluation and ranking.

## 8 Conclusions

In this paper, we explore the limitations of existing vision-language model (VLM) benchmarks, and introduce CHIRP, a novel benchmark designed to address these shortcomings. Our analysis reveals that a longer-form benchmark with open-ended questions quantifies multimodal understanding in ways that existing benchmarks do not. While current benchmarks evaluate contextually relevant responses, they often fail to capture the subtleties that humans value in long-form content.

**Expensive evaluations.** We acknowledge that generating and evaluating long-form responses, especially with human evaluators, can be resource-intensive. To mitigate this challenge, we have designed CHIRP to remain effective even at a smaller scale. Additionally, our findings suggest that AI evaluations can serve as a reliable proxy for human assessments, demonstrating similar overall trends in the same unique skill we aim to test for.

As VLMs continue to advance towards and beyond human-level performance on quantitative tasks, we emphasize the need to assess models on qualitative tasks that reflect the nuances of human preferences. Our work demonstrates that CHIRP is a viable benchmark for evaluating skills that have not been previously reported.

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

# A  Model training setup

## A.1  Process and data

In order to maximize comparability between models, we train all of them with the same hyper-parameters and data. The training of our VLMs is broken down into two phases: pretraining and finetuning. During pretraining, only the MLP projection is unfrozen, with both vision and language models frozen. The dataset used for this step is the *LLaVA Visual Instruct Pretrain LCS-558K* Liu et al. (2023a), which is a subset of the LAION/CC/SBU dataset, filtered with a more balanced concept coverage distribution. Following this, we do a finetuning step where we tuned all three components: the MLP projection, language model and vision encoder. The data that was used for this part of training is the LLaVA Visual Instruct 665K Liu et al. (2023a). This dataset contains 150K GPT-generated multimodal instruction-following data, in addition to using images from the Coco 2017 dataset Lin et al. (2015), the GQA dataseet Hudson & Manning (2019), the OCR-VQA dataset Mishra et al. (2019), the TextVQA dataset Singh et al. (2019), and the VisualGenome dataset Krishna et al. (2016).

Table 3: Parameter counts of the different CLIP VEs used. The largest CLIP model chosen is indeed "g" and not "big G".

| Model | Parameter Count |
|---|---|
| CLIP ViT B | 86 million |
| CLIP ViT L | 307 million |
| CLIP ViT H | 632 million |
| CLIP ViT g | 1 billion |

In the LLaVA 1.5 model release Liu et al. (2023a), the authors showed that when doing the finetuning of the language model, there was little difference between doing a full finetuning as opposed to a Low-Rank Adaptation (LoRA) Hu et al. (2021) finetuning. Therefore we trained all our models using a LoRA finetuning for the language model.

## A.2  Hyperparameters

Table 4a gives the hyper-parameters used for pretraining and Table 4b shows the hyperparameters used for finetuning all of the models. Due to different hardware being used to train different models, the gradient accumulation steps were changed for both the pretraining and finetuning steps in order to keep the batch size consistent between the different runs.

On a node consisting of 4 AMD Instinct MI250 Accelerators, pretraining would take about 4 hours and finetuning about 10 hours.

| Parameter | Value |
|---|---|
| Vision encoder | Frozen |
| Language model | Frozen |
| Projection learning rate | $10^{-3}$ |
| Use of fp16 | True |
| Projection type | mlp2x_gelu |
| Weight decay | 0 |
| Warmup ratio | 0.03 |
| Epochs | 1 |
| Batch size | 256 |

(a) For the pretraining

| Parameter | Value |
|---|---|
| Vision encoder learning rate | $5 \cdot 10^{-5}$ |
| Language model learning rate | $2 \cdot 10^{-5}$ |
| Projection learning rate | $2 \cdot 10^{-5}$ |
| Use of fp16 | True |
| Projection type | mlp2x_gelu |
| Weight decay | 0 |
| Warmup ratio | 0.03 |
| Epochs | 1 |
| Batch size | 128 |
| LoRA $r$ | 128 |
| LoRA $\alpha$ | 256 |

(b) For the finetuning

Table 4: Hyperparameters used during training

### A.3 Final loss plots

Plots showing the average loss of the last 10 iterations of training for each model of the Robin scaling suite.

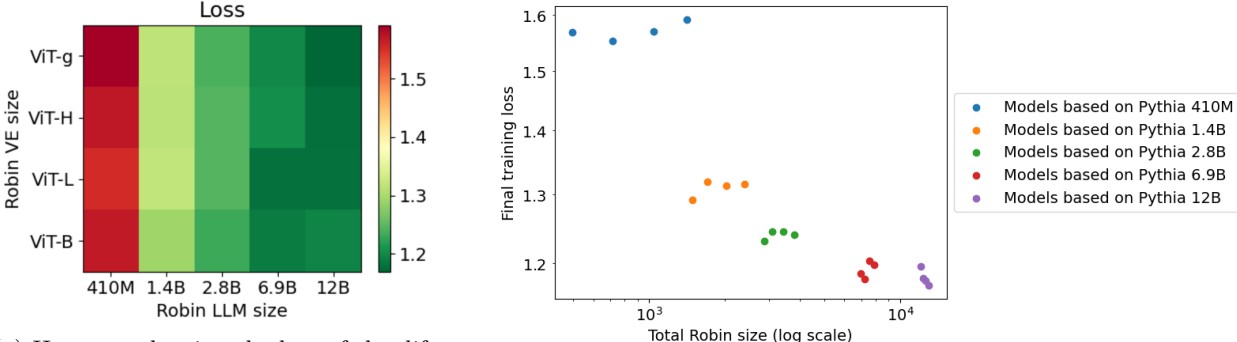

(a) Heatmap showing the loss of the different Robin models.

(b) Log-log plot showing how the loss scales with total parameter count.

Figure 7: Comparison of loss scaling and evaluation scores for the Robin models.

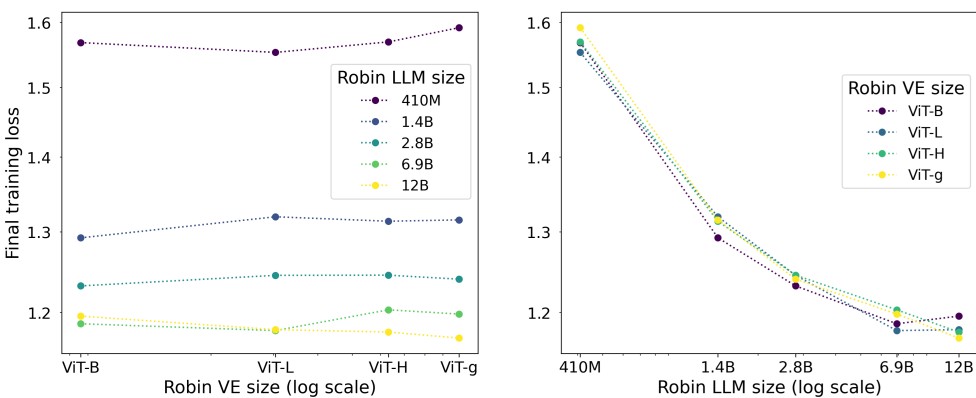

Figure 8: Log-log plots showing the scaling laws with *VE size* and *LLM size* respectively. The loss is calculated as an average over the last 10 iterations of training.

## A.4 Detailed benchmark scores of all the Robin scaling suite models

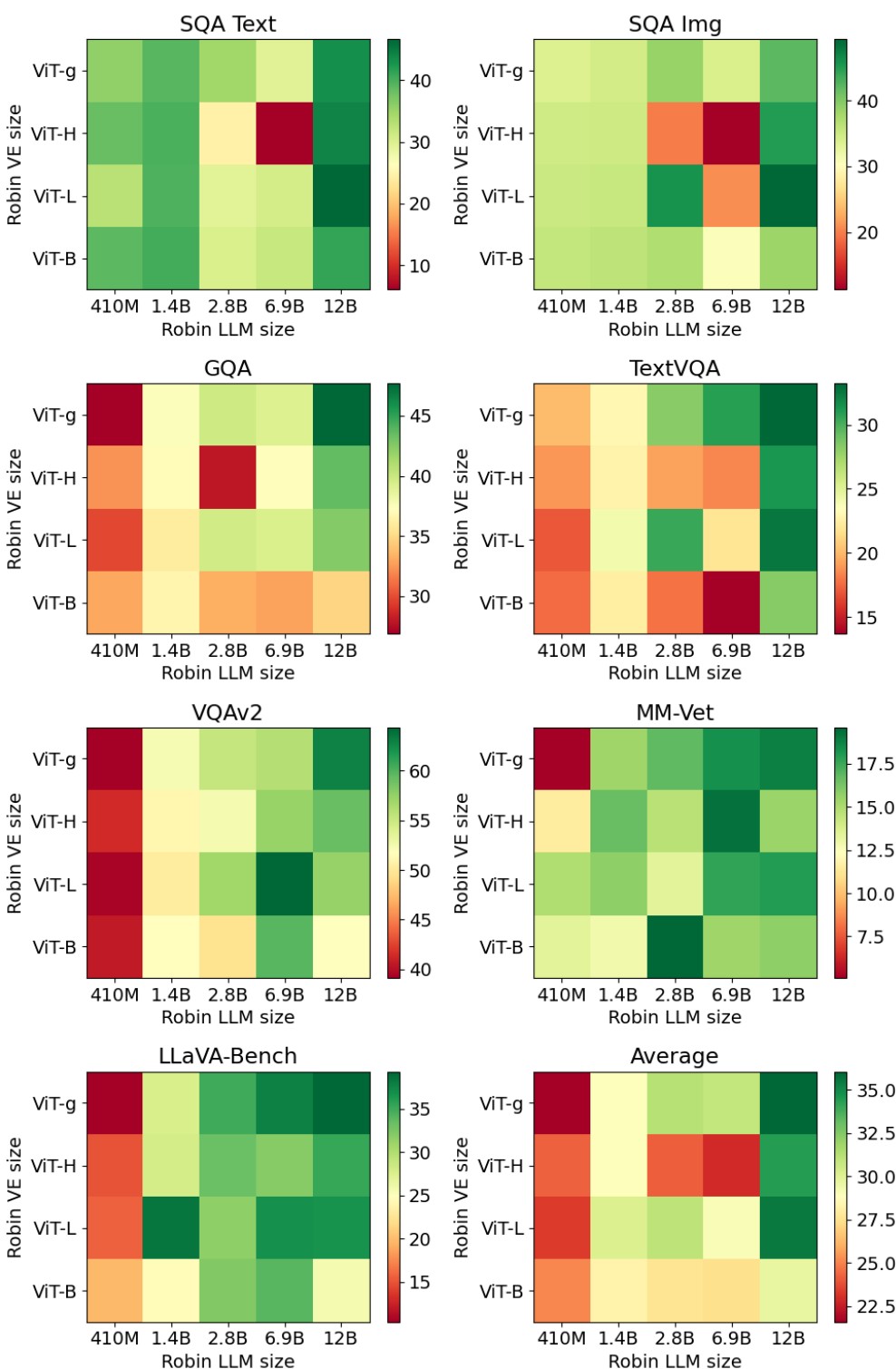

Figure 9: Heatmaps showing the performance of the different models of the scaling suite on the different benchmarks. For all graphs, higher is better.

Table 5: List of the results obtained by every model of the Robin scaling suite on the different benchmarks. Each model was run with LoRA finetuning for the LLM and unfrozen VE.

| LLM | VE | Link | SQA Text | SQA Img | GQA | TextVQA | VQAv2 | MM-Vet | LLaVA-Bench |
|---|---|---|---|---|---|---|---|---|---|
| Pythia 410M | CLIP ViT B 16 | link | 38.95% | 35.70% | 32.96% | 17.54% | 40.52% | 13.4% | 19.7% |
| Pythia 410M | CLIP ViT L 14 | link | 32.96% | 35.05% | 29.77% | 16.95% | 39.42% | 14.9% | 15.6% |
| Pythia 410M | CLIP ViT H 14 | link | 38.34% | 34.80% | 32.28% | 18.87% | 41.33% | 11.5% | 15.0% |
| Pythia 410M | CLIP ViT g 14 | link | 35.84% | 33.56% | 26.88% | 20.10% | 39.14% | 5.1% | 10.4% |
| Pythia 1.4B | CLIP ViT B 16 | link | 40.34% | 36.09% | 36.56% | 22.42% | 51.71% | 12.9% | 24.4% |
| Pythia 1.4B | CLIP ViT L 14 | link | 39.73% | 35.40% | 36.06% | 24.15% | 50.25% | 15.8% | 38.4% |
| Pythia 1.4B | CLIP ViT H 14 | link | 39.90% | 34.95% | 37.05% | 22.70% | 51.02% | 16.6% | 27.9% |
| Pythia 1.4B | CLIP ViT g 14 | link | 39.24% | 34.56% | 37.50% | 22.96% | 52.35% | 15.4% | 27.7% |
| Pythia 2.8B | CLIP ViT B 16 | link | 30.21% | 36.99% | 33.22% | 17.81% | 49.55% | 19.6% | 32.2% |
| Pythia 2.8B | CLIP ViT L 14 | link | 29.38% | 45.76% | 39.72% | 30.43% | 56.88% | 13.4% | 31.7% |
| Pythia 2.8B | CLIP ViT H 14 | link | 24.52% | 19.83% | 27.89% | 19.20% | 52.52% | 14.7% | 33.3% |
| Pythia 2.8B | CLIP ViT g 14 | link | 34.64% | 38.72% | 39.87% | 28.25% | 55.22% | 16.8% | 34.9% |
| Pythia 6.9B | CLIP ViT B 16 | link | 31.83% | 30.64% | 32.80% | 13.71% | 59.81% | 15.4% | 34.0% |
| Pythia 6.9B | CLIP ViT L 14 | link | 30.68% | 20.87% | 39.24% | 21.93% | 64.36% | 17.7% | 36.8% |
| Pythia 6.9B | CLIP ViT H 14 | link | 6.04% | 11.25% | 37.17% | 18.37% | 57.34% | 19.3% | 32.0% |
| Pythia 6.9B | CLIP ViT g 14 | link | 29.59% | 33.96% | 39.12% | 30.89% | 56.00% | 18.3% | 37.8% |
| Pythia 12B | CLIP ViT B 16 | link | 41.22% | 38.42% | 34.79% | 28.31% | 51.79% | 15.8% | 25.7% |
| Pythia 12B | CLIP ViT L 14 | link | 46.64% | 49.28% | 42.54% | 32.60% | 57.39% | 18.0% | 36.6% |
| Pythia 12B | CLIP ViT H 14 | link | 44.16% | 45.02% | 43.55% | 31.26% | 59.15% | 15.5% | 35.3% |
| Pythia 12B | CLIP ViT g 14 | link | 43.17% | 42.09% | 47.69% | 33.22% | 62.96% | 18.9% | 39.3% |

### A.5 Additional benchmarks

Table 6: List of the results obtained by every model of the Robin scaling suite.

| LLM | VE | OCR Bench | MathVista |
|---|---|---|---|
| Pythia 410M | CLIP ViT B 16 | 1.1% | 20.6% |
| Pythia 410M | CLIP ViT L 14 | 0.7% | 21.2% |
| Pythia 410M | CLIP ViT H 14 | 0.8% | 17.4% |
| Pythia 410M | CLIP ViT g 14 | 0.1% | 20.1% |
| Pythia 1.4B | CLIP ViT B 16 | 10.4% | 20.2% |
| Pythia 1.4B | CLIP ViT L 14 | 10.3% | 20.0% |
| Pythia 1.4B | CLIP ViT H 14 | 8.2% | 18.1% |
| Pythia 1.4B | CLIP ViT g 14 | 6.5% | 19.9% |
| Pythia 2.8B | CLIP ViT B 16 | 8.7% | 20.0% |
| Pythia 2.8B | CLIP ViT L 14 | 0.7% | 17.7% |
| Pythia 2.8B | CLIP ViT H 14 | 4.1% | 17.3% |
| Pythia 2.8B | CLIP ViT g 14 | 6.1% | 19.2% |
| Pythia 6.9B | CLIP ViT B 16 | 10.2% | 21.8% |
| Pythia 6.9B | CLIP ViT L 14 | 16.7% | 23.1% |
| Pythia 6.9B | CLIP ViT H 14 | 10.5% | 17.0% |
| Pythia 6.9B | CLIP ViT g 14 | 9.2% | 20.2% |
| Pythia 12B | CLIP ViT B 16 | 1.6% | 19.7% |
| Pythia 12B | CLIP ViT L 14 | 10.6% | 19.5% |
| Pythia 12B | CLIP ViT H 14 | 9.1% | 20.1% |
| Pythia 12B | CLIP ViT g 14 | 7.9% | 20.4% |

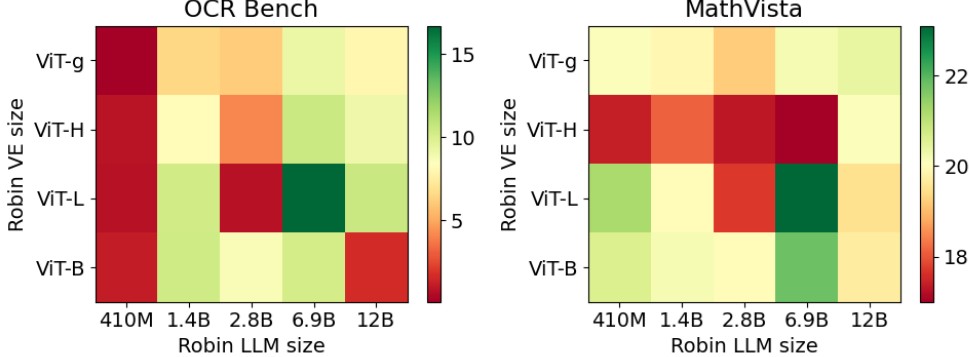

Figure 10: Heatmaps showing the performance of the different models of the scaling suite on the different benchmarks. For all graphs, higher is better.

# B  Further evaluations of the GQA and TextVQA prompts

## B.1  Graphs comparing the entire model suite on the different evaluation methods

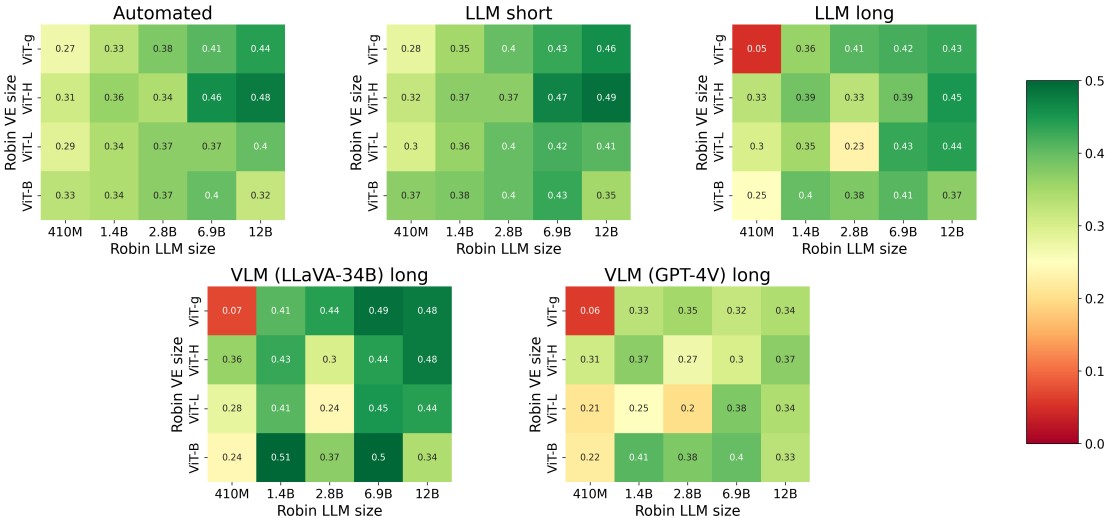

Figure 11: Accuracy of the Robin suite of models on the 100 GQA question sample calculated using different evaluation methods. Only weak scaling trends are apparent, irrespective of the evaluation method used.

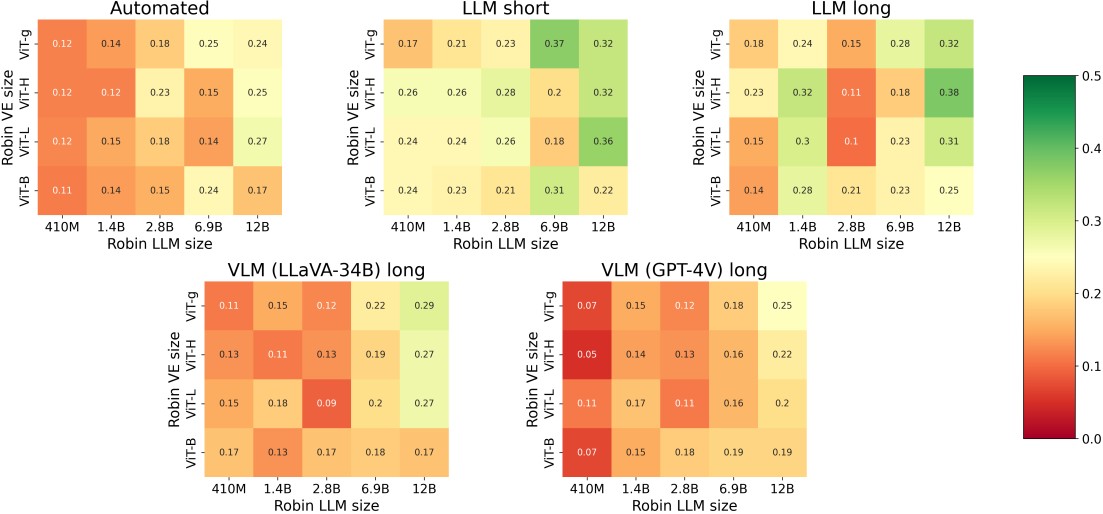

Figure 12: Accuracy of the Robin suite of models on the 100 TextVQA question sample calculated using different evaluation methods. Only weak scaling trends are apparent, irrespective of the evaluation method used.

## B.2 Verifying the results on an independent SoTA model

While other experiments in this paper are done using our Robin scaling suite with based on the Pythia LLMs, this section was done using LLaVA1.5-7B Liu et al. (2023a), in order to make sure that our results translated across models. We manually graded LLaVA1.5-7B responses on our sample of GQA questions. Using the known evaluations, we graded the accuracy of automated, LLM, and VLM evaluation methods. Results of grading accuracy on LLaVA1.5-7B are presented in the confusion matrix 13.

Some small, but important, usage details we noticed: The LLMs have a very low false positive rate, especially in contrast to their false negative rate. This suggests that for actual deployment, we could employ a two phase strategy, in which we assume the LLM is correct when it marks a long response as correct. When the LLM responds false, we fallback to a VLM. This strategy eliminates some VLM false negatives. The accuracy of this strategy is 88%, beating the other methods shown in Table 7.

We tried this strategy on our Robin models across the *LLM size* ablation. The results on GQA and textVQA are presented in Figure 14. Our results indicate that the joint LLM and VLM strategy provides a risk averse method of evaluation. Regardless of if the LLM or VLM evaluations are more accurate, the combined method provides a middle ground evaluation which performs slightly better on benchmarks where the LLM evaluates well, as GQA, but poorly when the VLM evaluation consistently outperforms the LLM evaluation, as in TextVQA.

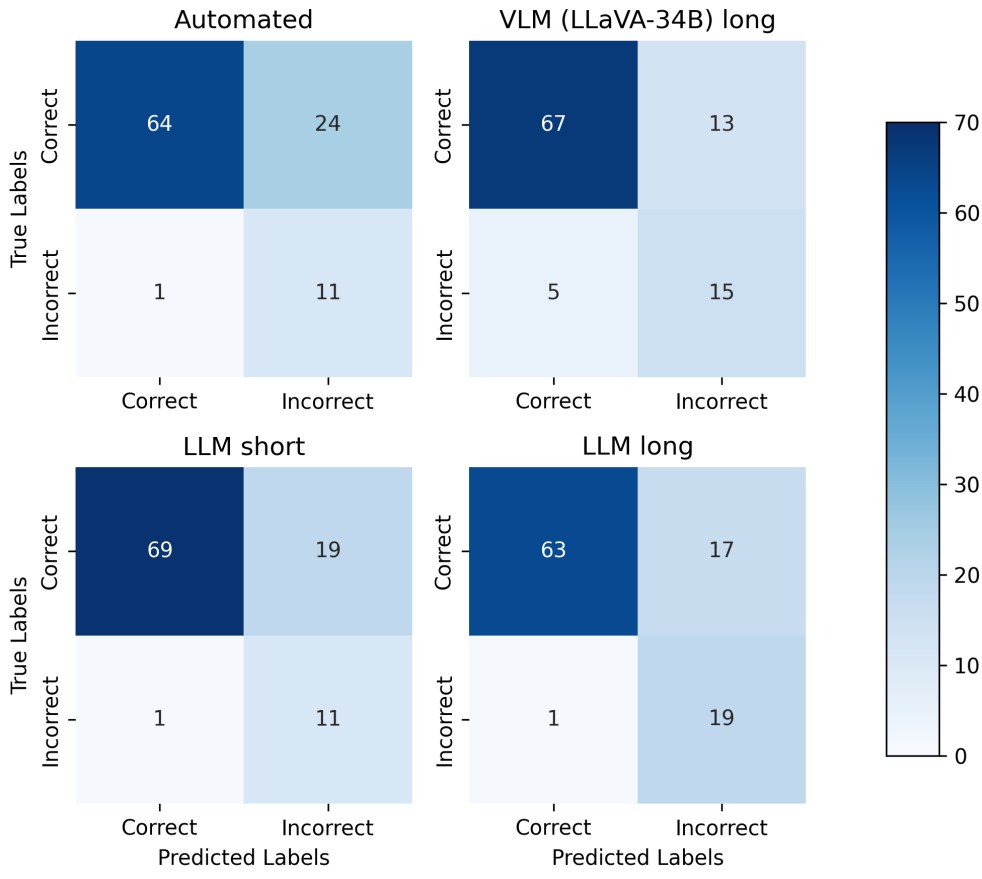

Figure 13: Confusion matrices of the different evaluation methods on the LLaVA1.5-7B responses.

Table 7: Accuracy of the different evaluation methods on the LLaVA1.5-7B responses.

| Method | Accuracy |
|---|---|
| string matching | 75% |
| LLM on short responses | 80% |
| LLM on long responses | 82% |
| VLM on long responses | 82% |
| Join LLM+VLM evaluation | 88% |

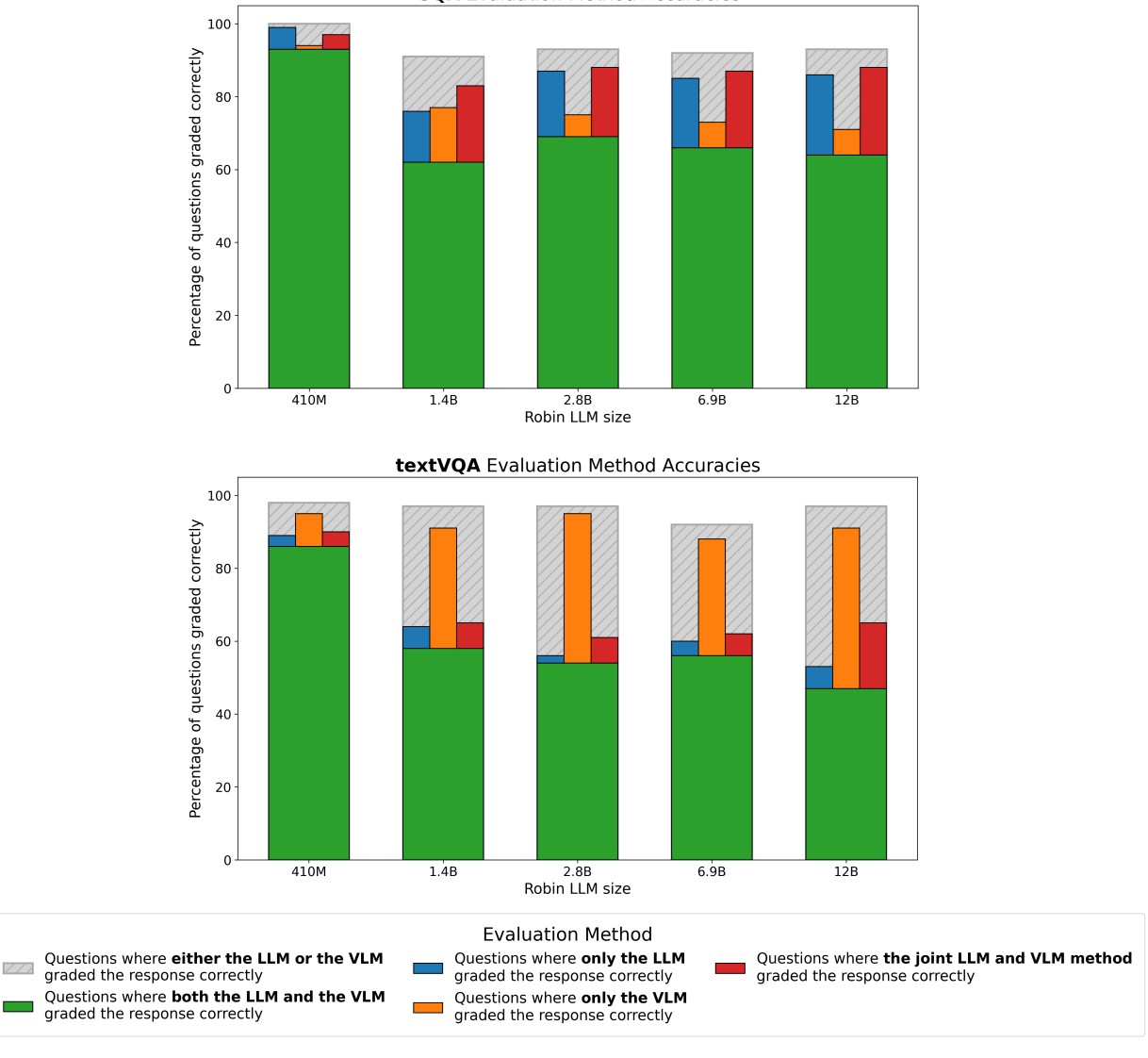

Figure 14: Accuracy of different evaluation methods on a sample of 100 questions from both GQA and textVQA.

### B.3 Prompts used for the AI evaluations

```
messages = [
{"role": "system", "content": "You will be provided with a question about some
↪  image, the correct answer to the question, and a students response. Grade
↪  whether or not the student answered the question correctly based on the correct
↪  answer that is provided. Respond correct, or incorrect, depending on the given
↪  response."},
{"role": "user", "content": f"Question: {question}\n\nCorrect Answer:
↪  {ground_truth}\n\nStudents Answer: {vlm_response}"}
]
```

Figure 15: Prompt passed to GPT-4 for the LLM evaluation of both long and short responses on GQA and textVQA

```
messages = [
{"role": "user", "content": <IMAGE> + f"{question}"}
{"role": "llava", "content": LLaVAs response}
{"role": "user", "content": f"Based on your answer, grade the following response to
↪  the same question as correct or incorrect.\n\nResponse: {response}"}
]
```

Figure 16: Prompt used for LLaVA-34B evaluation of both long and short responses on GQA and textVQA. We first asked LLaVA-34B to answer the question, then asked it to evaluate the models response taking into account its own response.

```
instruction = """
You are a helpful assisstant. You will be shown an image and a related question, along with a response from an
↪  assistant. The assistants' responses are meant to answer the given question.

Your task is to evaluate the response to the given question about the image.

Image:
"""

response_eval = f"""
Question: {question}

Assistants Response: {response}

Please evaluate whether this response is correct or not. You can mark questions that include false details about the
↪  image as incorrect. First reason about your thought process before giving the final answer.
"""
gpt_response = openai_client(
    model = "gpt-4-vision-preview",
    messages=[
      {
        "role": "user",
        "content": [
          {"type": "text", "text": instruction},
          {
            "type": "image_url",
            "image_url": {
              "url": image_url,
            },
          },
          {"type": "text", "text": response_eval},
        ],
      }
])

final_evaluation = openai_client(
    model="gpt-3.5-turbo",
    messages=[
      {
        "role": "user",
        "content": f"You will receive an evaluation of an assistant's response to a question. Your task is to analyze
        ↪  the text, and determine whether the assistants response was correct or incorrect. Please only respond
        ↪  with the word \"Correct\" or \"Incorrect\". If the response is partially correct, you may respond with the
        ↪  phrase \"Partially Correct\". \n\nEvaluation:\n{response}"
      }
    ]
)
```

Figure 17: Prompt used for GPT-4V evaluation of long responses on GQA and textVQA. We first asked GPT-4V to evaluate the question answer pair and reason about its answer. We then asked GPT-3.5 to parse the final answer. "Partially Correct" results were treated as incorrect.

### B.4 Qualitative examples

### B.4.1 Examples of issues in the GQA dataset

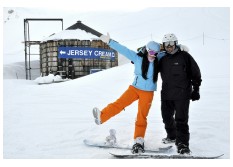 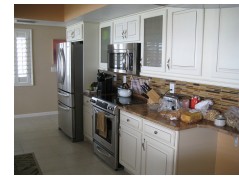 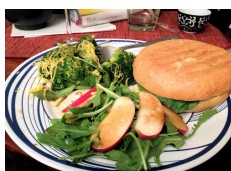

| **Question:** | What device is behind the man? | Is the stove to the left of a drawer? | Is there a cup near the plate? |
|---|---|---|---|
| **Ground truth:** | The device is a television. | No, the stove is to the left of a toaster. | No, there is a mat near the plate. |

Table 8: Examples of GQA questions from our sample that have incorrect ground truth answers.

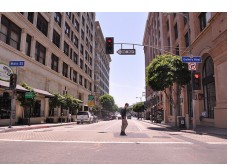

| **Question:** | What sits next to the street that is made of asphalt? | What is on the motorbike that the person is riding? | Does the window look square? |
|---|---|---|---|
| **Ground truth:** | The signal light sits next to the street. | The mirror is on the motorbike. | Yes, the window is square. |

Table 9: Examples of ambiguous GQA questions from our sample which have multiple potential correct answers.

### B.4.2 Examples of issues in the TextVQA dataset

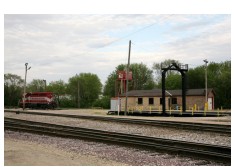 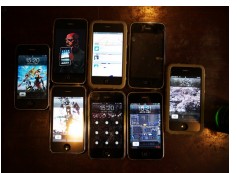

| **Question:** | This railway track? | Does the parking has more space? | 15:20  15:21  15:20 15:20 15:21? |
|---|---|---|---|
| **Ground truth:** | yes; no; unanswerable; not a question; ... | yes; unanswerable; answering does not require reading the text in the image; ... | yes; not a question; unanswerable; ... |

Table 10: Examples of ambiguous TextVQA questions from our sample. The TextVQA dataset provides 10 ground truth answers per question, seperated here by ";".

### B.4.3 Examples of the grading disagreement between methods

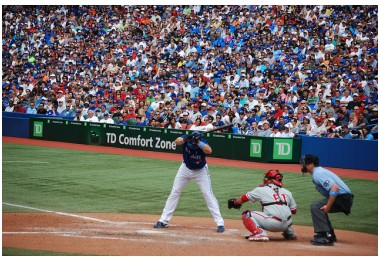

| **Question:** | Who is wearing the helmet? |
|---|---|
| **Ground truth:** | the batter is wearing the helmet |
| **LLaVA-1.5-7B:** | the player |
| **GQA evaluation:** | Incorrect |
| **LLM evaluation:** | Correct |

Table 11: A sample where the LLM marked the response correctly but the model response does not contain a direct string match to the ground truth answer, and thus the automated evaluation fails.

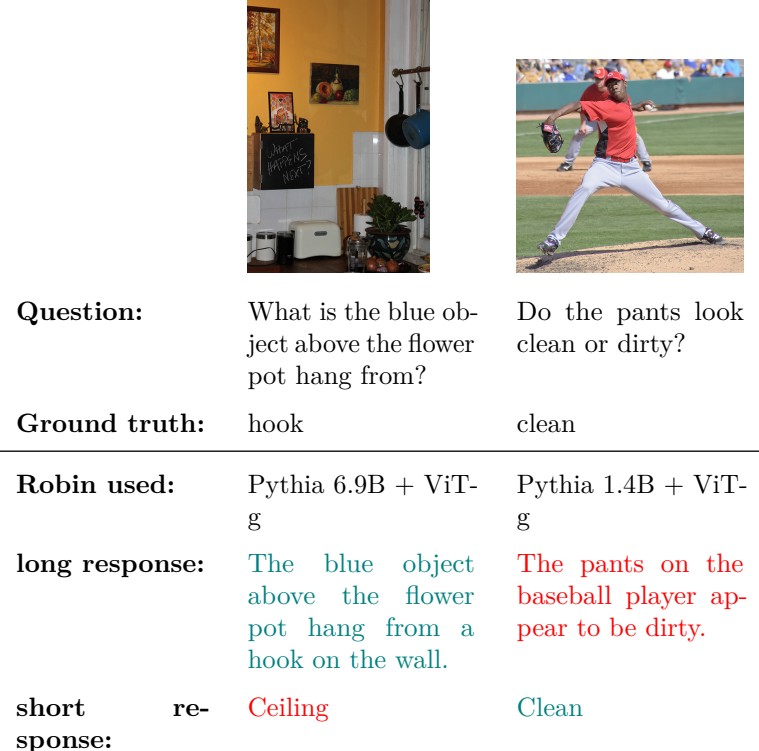

| | | |
|---|---|---|
| **Question:** | What is the blue object above the flower pot hang from? | Do the pants look clean or dirty? |
| **Ground truth:** | hook | clean |
| **Robin used:** | Pythia 6.9B + ViT-g | Pythia 1.4B + ViT-g |
| **long response:** | The blue object above the flower pot hang from a hook on the wall. | The pants on the baseball player appear to be dirty. |
| **short response:** | Ceiling | Clean |

Table 12: Examples of response differences when prompting for short vs. long responses.

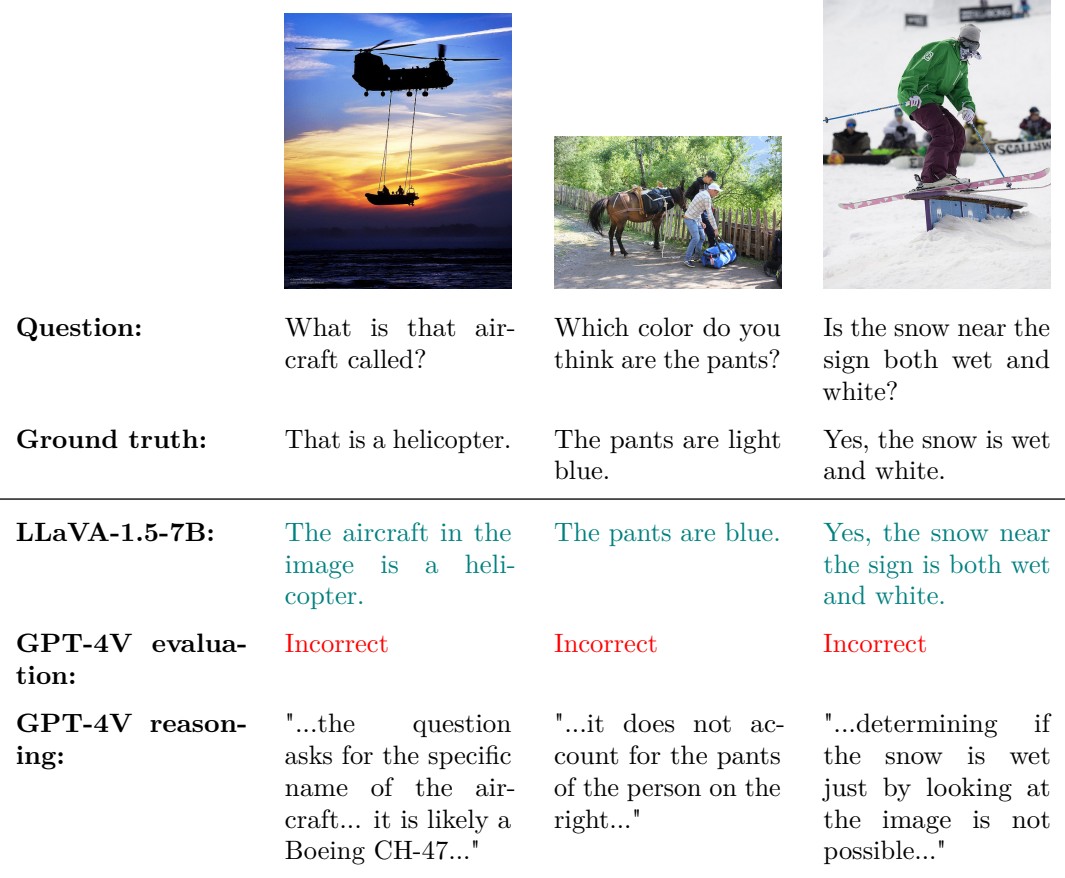

| | | | |
|---|---|---|---|
| **Question:** | What is that aircraft called? | Which color do you think are the pants? | Is the snow near the sign both wet and white? |
| **Ground truth:** | That is a helicopter. | The pants are light blue. | Yes, the snow is wet and white. |
| **LLaVA-1.5-7B:** | The aircraft in the image is a helicopter. | The pants are blue. | Yes, the snow near the sign is both wet and white. |
| **GPT-4V evaluation:** | Incorrect | Incorrect | Incorrect |
| **GPT-4V reasoning:** | "...the question asks for the specific name of the aircraft... it is likely a Boeing CH-47..." | "...it does not account for the pants of the person on the right..." | "...determining if the snow is wet just by looking at the image is not possible..." |

Table 13: A few examples of how GPT-4V evaluations are stricter than ground truth or most human evaluators.

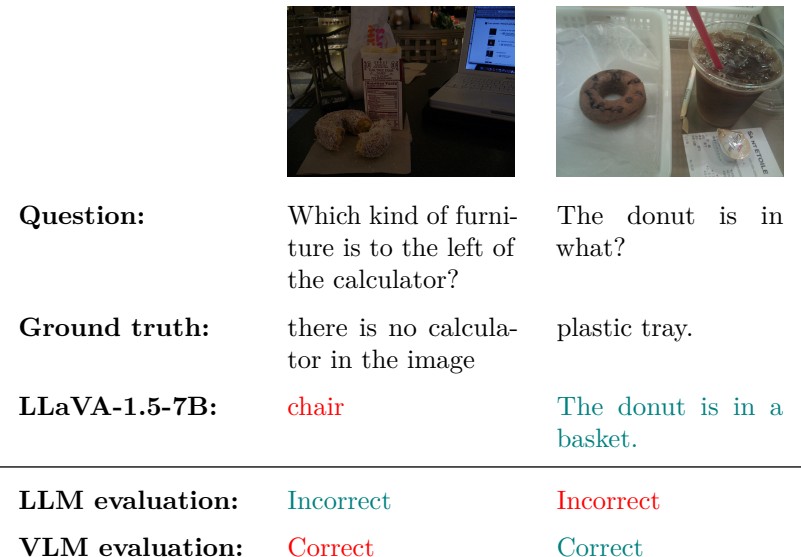

| | | |
|---|---|---|
| **Question:** | Which kind of furniture is to the left of the calculator? | The donut is in what? |
| **Ground truth:** | there is no calculator in the image | plastic tray. |
| **LLaVA-1.5-7B:** | chair | The donut is in a basket. |
| **LLM evaluation:** | Incorrect | Incorrect |
| **VLM evaluation:** | Correct | Correct |

Table 14: Examples of VLM and LLM evaluations of GQA questions. On the left, the VLM is more likely to hallucinate and agree with a trick question's given answer than an LLM or automated evaluator. On the right, the VLM shows greater flexibility in accepting alternative correct answers.

# C CHIRP

Benchmark questions and images available at `https://huggingface.co/datasets/Anonymous1234565/CHIRP`

## C.1 Benchmark Details

**Question Categories.** We identified 8 distinct categories of questions that demand comprehensive image analysis. For each category, we prompted GPT-4 to come up with questions and corresponding image descriptions. After refining these by hand, we pass the image descriptions to Dall-E 3 to generate the described image. We will aslo iterate the description untill obtaining an image of high quality. We present the distribution of questions across different categories in Figure 18. The exact categories and their explanations are as follow:

- **Descriptive Analysis:** This category involves questions that test the model's ability to identify and describe the physical elements in an image, including color, position, and interaction and also to recognize specific details.

- **Inferential Reasoning:** It examines the model's ability to infer things from the image, including predicting possible subsequent events, making assumptions about previous contexts, and hypothesizing alternative scenarios that contradict the present one in the image.

- **Contextual Understanding:** This category tests the model's awareness of the importance of context in image comprehension. This might involve understanding geographical or temporal aspects that bear upon the image.

- **Emotional and Psychological Understanding:** It measures the model's ability to gauge emotions and psychological states from an image. This incorporates interpreting the visible emotional expressions of characters in the image and hypothesizing about their mental state.

- **Ethical Evaluations:** Questions in this category check how the model deals with the ethical implications of images. Can it recognize potential ethical concerns and judge the public display acceptability of an image with respect to generally accepted ethical guidelines?

- **Abstract Understanding:** These questions gauge the model's capacity for abstract thought — can it identify underlying themes or messages in the image that aren't immediately visible? Can it engage in philosophical interpretation?

- **Creative and Subjective Analysis:** This category gauges the model's creativity and its ability to express subjective views on the image. It includes crafting extended narratives based on the image scenery and presenting a personal point of view for the image.

- **Visual Aesthetics Evaluation:** This category examines the model's ability to evaluate the visual aesthetics of an image including aspects like balance, symmetry, colour composition, lighting, etc.

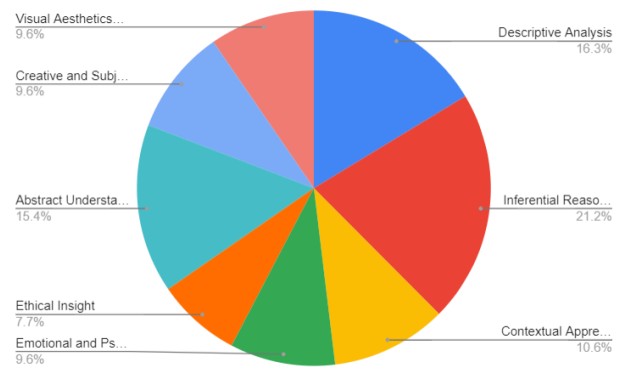

Figure 18: CHIRP single question category distribution.

### C.2 Evaluation procedure

### C.2.1 Evaluation Criteria

We evaluated pairwise comparison of responses on each of the following criteria:

- **Overall Preference:** Which assistant's response do you prefer overall, considering all factors?
- **Relevance and Completeness Evaluation:** Which assistant's response is more relevant to the question and provides a more complete answer?
- **Understanding and Reasoning:** Which assistant's answer displays a better understanding of concepts and better reasoning in its response?
- **Hallucination Evaluation:** Which assistant accurately describes the image without adding or describing objects or elements that don't exist in the image?
- **Detail Evaluation:** Which assistant's description of the image is more detailed, taking into consideration both the amount and quality of the details provided?

### C.2.2 Human Survey

We use Cloud Research to conduct human evaluations of our models on the CHIRP benchmark. We conducted 3 different studies on Cloud Research:

1. **LLM Size Study.** A study comparing our ViT-g VE models accross the 5 LLM sizes.

2. **VE Size Study.** A study comparing our 12b parameter LLM models accross the 4 VE sizes.

3. **All Robin Models Study.** A study comprising of matchups comparing all 20 of our models.

For each of the 104 questions, we randomly sampled a portion of all possible pairwise combinations of models involved in the study.

For the *LLM size* study, 5 of the 10 combinations of matchups between models were randomly sampled for each question. For each of those sampled matchups, we asked participants to indicate their preference for the 5 evaluation criteria. We only allowed each participant to respond with their preferences for a single matchup. This led to a total of 520 individual responses (104 questions * 5 matchups). A breakdown of the questions asked in each survey is presented in Table 15. An example of the participant interface is shown in Figure 19.

Table 15: Breakdown of CHIRP human survey evaluation matchups.
* For All Models, evaluators were only asked to provide preferences for the overall category.

| Survey | CHIRP questions | Total Matchups | Matchups Sampled | Participants | Criteria Evaluated |
|---|---|---|---|---|---|
| **LLM Size** | 104 | 10 | 5 | 520 | 5 |
| **VE Size** | 104 | 6 | 3 | 312 | 5 |
| **All Models** | 104 | 190 | 25 | 2600 | 1[*] |

For all surveys, we targeted English-speaking participants aged 18-50 who had graduated high school. Participants were compensated at an estimated rate of \$0.10 per minute, following Cloud Research guidelines. The first two studies, which required evaluating five criteria, were estimated to take 2 minutes each, with a compensation of \$0.20. The last study, requiring evaluation of one criterion, was estimated to take 1 minute, with a compensation of \$0.10. This rate ensured that participants were paid at least minimum wage. Post-study analysis showed that average response times agreed with our estimates, confirming compliance with minimum wage requirements.

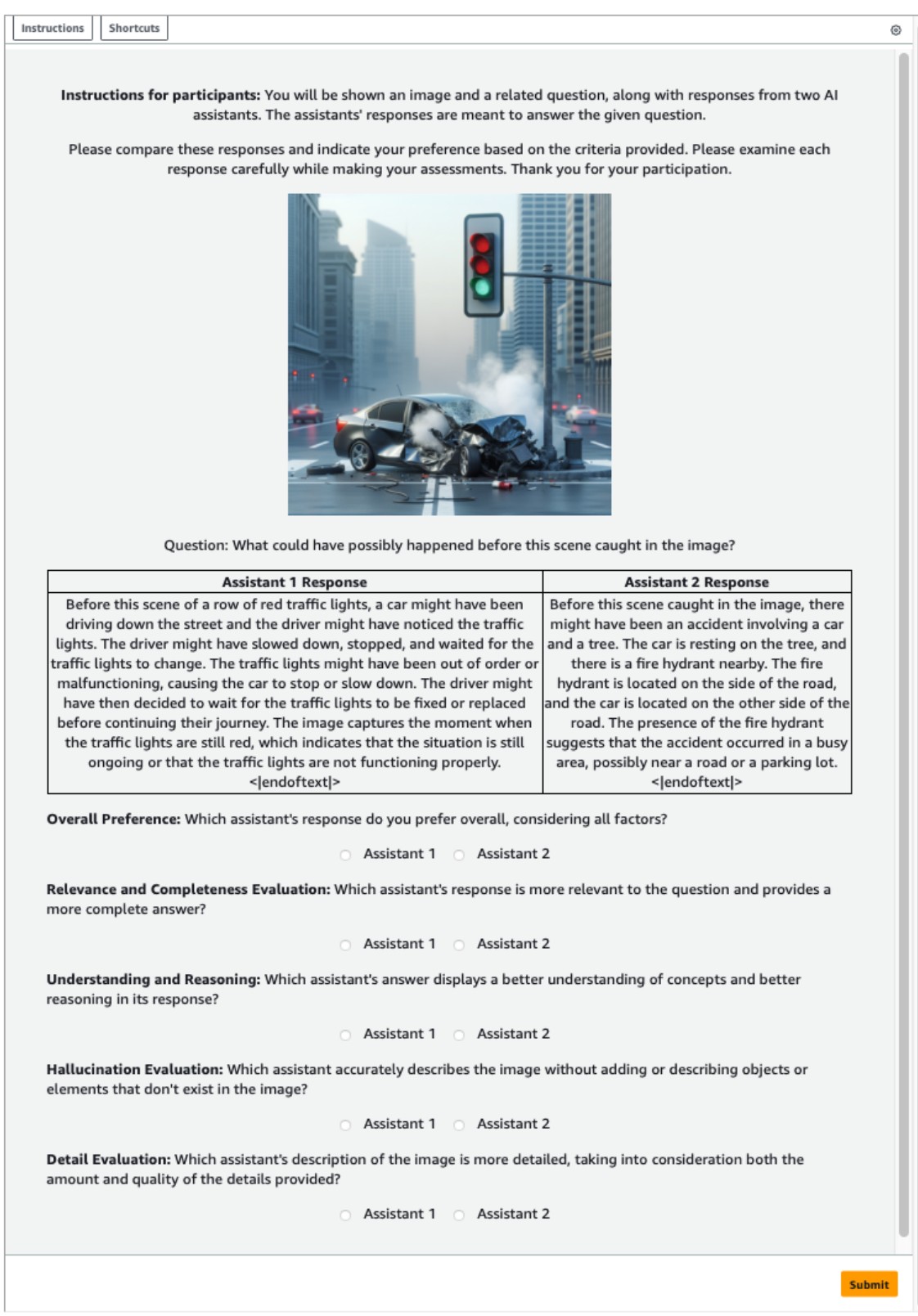

Figure 19: Example of survey questions displayed to a human evaluator on Cloud Research. The same instructions and format were used for all studies conducted on Cloud Research.

### C.2.3 AI evaluations

AI evaluations were run on all 10 matchups for both the *LLM size* study and *VE size* study. For the *all Robin models* study, we only ran **GPT-4V (R)** on a sample of 50 of the 190 possible matchups for each question. The prompts used for the evaluation are shown in Figures 20 and 21.

```
category_prompts = [
      "Which response do you prefer overall, considering all factors?",
      "Which response is more relevant to the question and provides a more complete
      ↪  answer?",
      "Which response displays a better understanding of concepts and better
      ↪  reasoning?",
      "Which response more accurately describes the image without adding or
      ↪  describing objects or elements that don't exist in the image?",
      "Which response's description of the image is more detailed, taking into
      ↪  consideration both the amount and quality of the details provided?"
]

response_eval = f"""
Here are two responses to the same question:

Response 1: {response_1}
Response 2: {response_2}

{category_prompts[category]}
Respond with the number 1 or 2 corresponding to the better answer.
"""

messages = [
{"role": "user", "content": <IMAGE> + f"{question}"}
{"role": "llava", "content": LLaVAs_response}
{"role": "user", "content": response_eval}
]
```

Figure 20: Prompt used for LLaVA-34B evaluation of model responses on the CHIRP benchmark.

```python
instruction = """
You are a helpful assisstant. You will be shown an image and a related question, along with responses from two
↪  assistants. The assistants' responses are meant to answer the given question.

Your task is to compare and evaluate the two responses to the given question about the image.

Image:
"""
categories = [
        "Which assistant's response do you prefer overall, considering all factors?",
        "Which assistant's response is more relevant to the question and provides a more complete answer?",
        "Which assistant's response displays a better understanding of concepts and better reasoning?",
        "Which assistant accurately describes the image without adding or describing objects or elements that don't
        ↪  exist in the image?",
        "Which assistant's description of the image is more detailed, taking into consideration both the amount and
        ↪  quality of the details provided?"
]

response_eval = f"""
Question: {question}

Assistant 1 Response: {response_1}

Assistant 2 Response: {response_2}

{categories[category]}
Please do not provide Tie as an evaluation. You have to select between Assistant 1 or Assistant 2. {"Reason about
↪  your thought process before giving the final answer." if reasoning else "Please respond with only the number
↪  corresponding to the assistant with the preferred response."}
"""
gpt_response = openai_client(
    model = "gpt-4-vision-preview",
    messages=[
      {
        "role": "user",
        "content": [
          {"type": "text", "text": instruction},
          {
            "type": "image_url",
            "image_url": {
              "url": image_url,
            },
          },
          {"type": "text", "text": response_eval},
        ],
      }
])

if not reasoning:
    return gpt_response

final_evaluation = openai_client(
    model="gpt-3.5-turbo",
    messages=[
      {
        "role": "user",
        "content": f"You will receive an evaluation of two responses including the preferred assistants response.
        ↪  Your task is to analyze the text, determine which assistant's response is preferred, and output the number
        ↪  corresponding to the preferred assistant (either 1 or 2). Please only respond with the number
        ↪  correspondig to the preferred assistant and no additional information. For exmaple: 2.
        ↪  \n\nEvaluation:\n{gpt_response}"
      }
    ]
)

return final_evaluation
```

Figure 21: Prompts used for GPT-4V evaluation of model responses on the CHIRP benchmark. The *reasoning* variable in the psuedocode indicates whether the **GPT-4v (R)** (reasoning) or **GPT-4v (S)** (simple) prompt is used. In the case of **GPT-4v (R)** prompts, the final choice is extracted using GPT-3.5.

### C.2.4 Elo Score Graphs

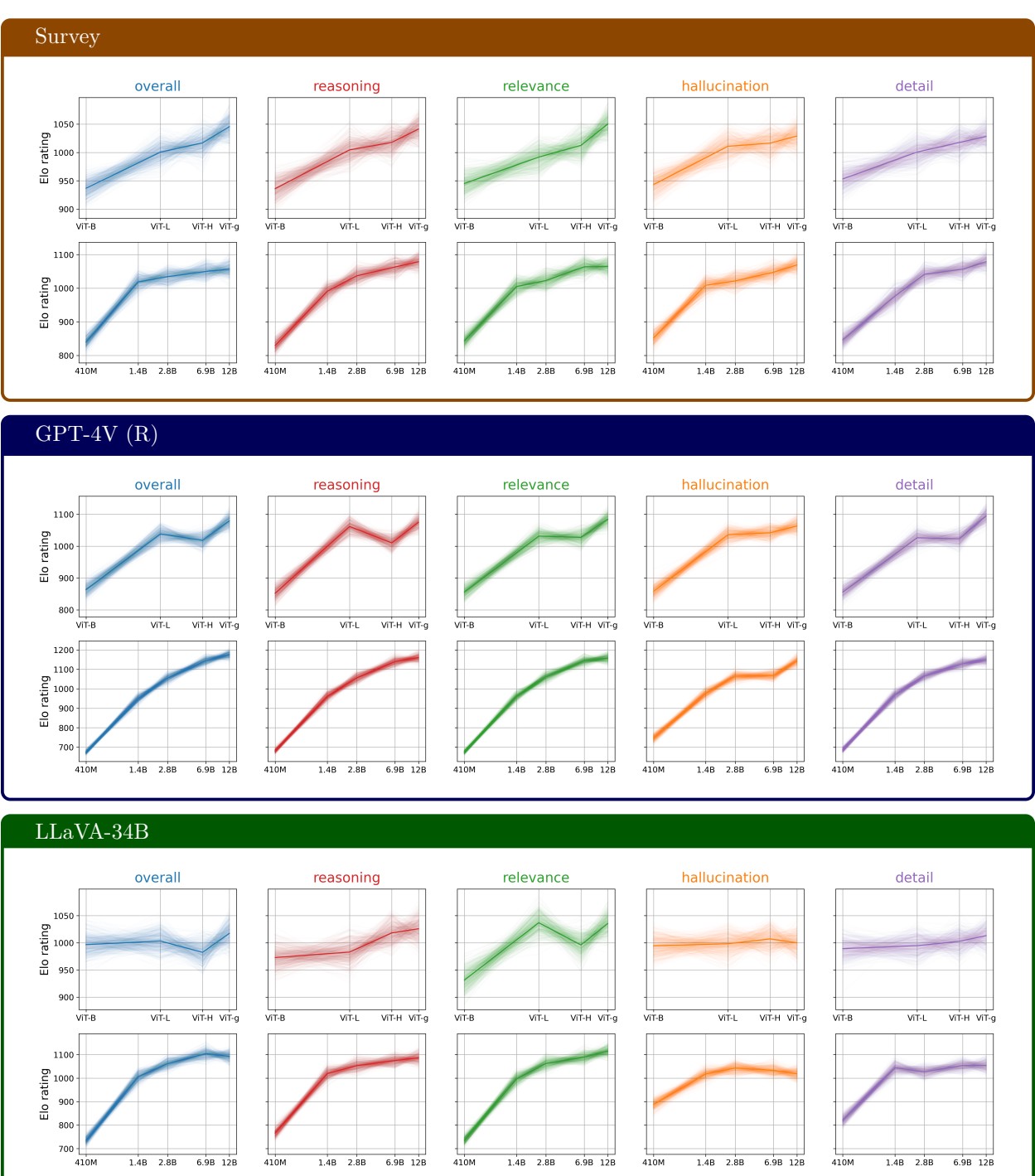

Figure 22: Elo scores calculated over LLM size and VE size on the 5 different evaluation criteria of CHIRP using the 3 different evaluators. Graphs are calculated using bootstrapping on 500 samples. Each sample is drawn with low transparency and the solid lines indicate the mean over samples for the respective category. For each evaluator, the first row of graphs concerns the *VE size* ablation and the second row concerns the *LLM size* ablation, with all X-axis being the size in log scale.

### C.2.5 Cohen's Kappa Calculation

We calculate Cohen's Kappa to compare AI evaluations with human evaluations, whilst accounting for the random chance of agreement. We compare only the matchups that were sampled in the human surveys against the AI evaluations of those same matchups. Cohen's Kappa ($\kappa$) is calculated according to the formula:

$$\kappa = \frac{p_o - p_e}{1 - p_e}$$

$p_o$ is the relative agreement: the proportion of matchups where the different evaluators agree on their preference.

$p_e$ is the hypothetical probability of chance agreement: we calculate this term for all combinations of matchups separately. Let $p_{e_{(a,b)}}$ be the probability $p_e$ for an individual matchup $(a,b)$. Namely, for each matchup of models $a$ and $b$, and an evaluator $\mathcal{E}$, the proportion of matchups where model $a$ is preferred will be represented as $p_{\mathcal{E}}(a|a,b)$. The probability of two evaluators $\mathcal{E}$ and $\mathcal{F}$ agreeing randomly for a matchup $(a,b)$ is then:

$$p_{e_{(a,b)}} = p_{\mathcal{E}}(a|a,b) * p_{\mathcal{F}}(a|a,b) + p_{\mathcal{E}}(b|a,b) * p_{\mathcal{F}}(b|a,b)$$

$p_e$ is then calculated by taking the weighted average over the frequency of matchups $f_{(a,b)}$ present in the survey:

$$p_e = \sum_{(a,b)} f_{(a,b)} * p_{e_{(a,b)}}$$

### C.2.6 CHIRP and training loss correlation

In the study with all the Robin models, comprising of matchups comparing all 20 of our models, we examine the extent to which evaluation methods tend to favor the model with the lower average training loss. In Table 16 we show the percentages of matchups where the evaluator choses the model with the lowest loss. The lowest loss used is the average of the loss in the final 10 steps of training in order to smooth out the spikes.

Furthermore, we compute the distance correlation Székely et al. (2007) (dCor) between the Elo scores and the model loss. This distance correlation is shown in Table 17 and captures both linear and non-linear associations between two vectors.

Our analysis reveals that both human surveys and GPT-4V (R) are highly correlated to the model training loss. This indicates that the training loss remains a good first estimator of the performance of a model on this benchmark, as in LLMs Kaplan et al. (2020); Ru et al. (2020). Furthermore, GPT-4V (R) correlates particularly well with the model training loss. This could be due to different factors such as the higher variance in the responses which is intrinsic to human evaluations and deserves to be explored further in future research. The lower alignment portrayed by Table 16 of the decorrelation of the model loss and parameter count in the larger models, based on Pythia 6.9B and 12B, as well as the loss for different VEs on a give LLM being rather grouped, as shown in Appendix A.3.

| Method | Agreement |
|---|---|
| Human Survey | 60.8% |
| GPT-4V (R) | 71.2% |

Table 16: Percentage of time the evaluators preferred the model with lower training loss.

| Method | dCor |
|---|---|
| Human Survey | 0.91 |
| GPT-4V (R) | 0.96 |

Table 17: Distance correlation between the models' Elo scores and training loss.

### C.2.7 Logits Agreement

To ensure that GPT-4V (R) evaluations of CHIRP considers the information from the images, we also calculate the response's text token probabilities. Using OpenAI's Davinci-002 model OpenAI (2024), we determine the

average log probability of tokens in each model's response to CHIRP questions. In the study comprising of matchups from all 20 of our Robin models, the highest log probability responses and the GPT-4V (R) evaluated responses agreed 48.7% of the time. This is practically equivalent to random chance agreement, which would be at 50%. This near-random agreement suggests that VLM evaluations are considering factors beyond the probability of response words occurring together and are indeed investigating the image.

### C.2.8 Further Explanations on Contradictions

Because human surveys were limited to 5 pairwise comparisons per question per category, we only calculate contradictions using those same 5 comparisons in AI evaluations. In order to evaluate logical contradictions, we start by building a directed graph where each model is a node and the link between 2 nodes is the user preference. For instance, if the model based on Pythia 6.9B is preferred over the model based on Pythia 2.8B, there will be a directional link from the Pythia 2.8B based model to the Pythia 6.9B based model. A logical contradiction is when a cycle is created in the graph.

Figure 23 illustrated this, with a contradiction in sub-figure **a** as users indicated they preferred the model based on Pythia 6.9B, over the one based on Pythia 1.4B, over the one based on Pythia 410M, which implies that the model based on Pythia 6.9B should be preferred over the model based on Pythia 410M. However, human evaluations showed that the model based on Pythia 410M is preferred over the one on Pythia 6.9B, hence the contradiction.

Note that contradictions themselves have nothing to do with the sizes of the models, but rather if there was an inconsistency in the transitivity of preferences for a given question.

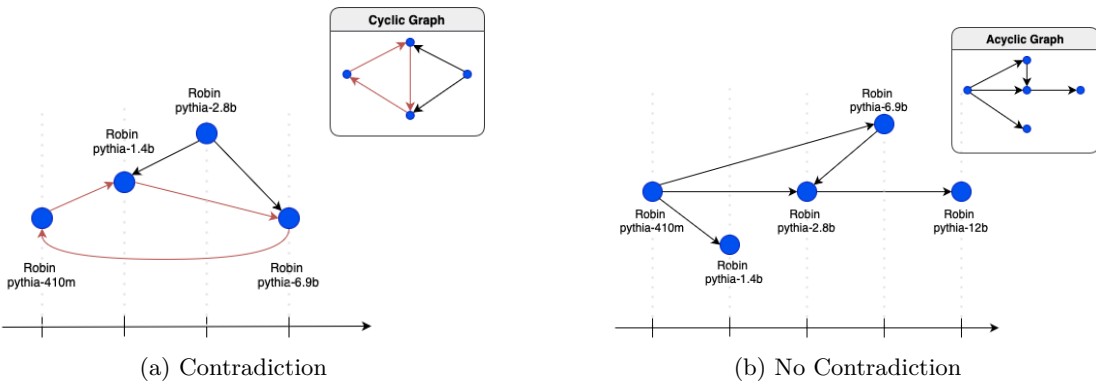

(a) Contradiction                         (b) No Contradiction

Figure 23: Visualization of model preferences over multiple human evaluators for a single question. Arrows point toward the evaluators preferred model. Contradictions take the form of cycles in the graph. **Left.** Example of a contradiction in preferences. **Right.** Example where preferences remain consistent.

### C.2.9  Graphing the results of CHIRP on the Robin suite

The graph in figure 24 shows the model preference of the human evaluators. Arrows from model A to B indicate that users preferred the outputs of model B over model A. Not all users had the same preference, therefore the stronger the arrow, the more a consensus was reached amongst the users on their preferred model. A weaker, more transparent, arrow indicates that the users were more divided on their preferred model, and that therefore this preference is less denoted. This can be seen as thicker arrows are more trustworthy. "Ties", where as many users answered in favor of one or the other model are not shown. We also note two main colors: the green arrows are for the user preferences which support our hypothesis that users prefer larger models, while the red arrows indicate user preferences that do not support this hypothesis. We see an overwhelming amount of preference for larger models, with a notable exception for models using the CLIP ViT-B vision encoder and Pythia 410M LLM, where this trend is reversed.

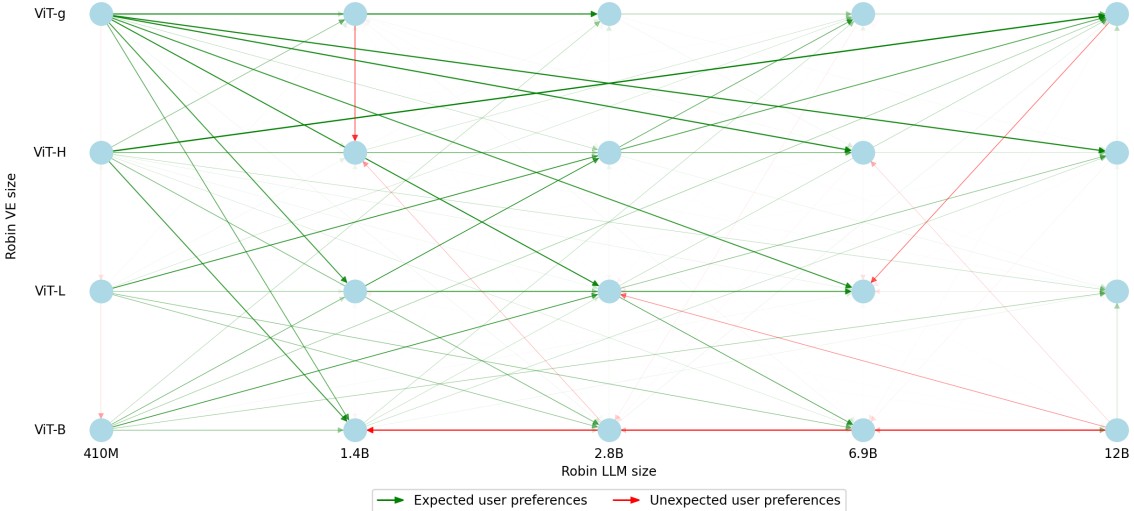

(a) Graph showing the complete user preferences in the "Overall" category of the Robin suite.

Figure 24: Visualization of model preferences over multiple human evaluators for the CHIRP benchmark. Arrows point toward the evaluators preferred model. The expected preference indicates when users preferred the larger model, while an unexpected preference denotes users preferring the smaller model.

