# OpenReview forum: "Robin: a Suite of Multi-Scale Vision-Language Models and the CHIRP Evaluation Benchmark"
_TMLR — Rejected by TMLR_

### Review · Reviewer_1XKB · 2025-03-01

**Summary Of Contributions:**

This paper systematically evaluates the limitations of commonly used VLM benchmarks, particularly in VQA-style datasets. To fairly assess these limitations, the authors introduce Robin, a family of VLM models trained with varying vision encoder and language decoder parameter sizes, following a design similar to LLaVA. Using Robin, the authors find that normalized VQA performance does not scale proportionally with model size. They identify three key issues causing this discrepancy:

1. Poorly designed string-matching evaluation in VQA benchmarks.
2. Incorrect ground-truth answers.
3. Variability in performance due to different prompting strategies.

To address these concerns, the paper proposes CHIRP, a new benchmark with 102 questions. Instead of traditional string-matching evaluation, CHIRP employs preference-based pairwise comparisons of long-form responses. The authors further automate evaluation by using powerful VLMs (e.g., GPT-4V and LLaVA-34B) as AI raters. Their findings suggest that AI-based preference ratings do not always align with human evaluations—particularly, LLaVA-34B provides highly consistent ratings across Robin variants, while GPT-4V [R] (a reasoning-focused model) exhibits stronger correlation with human judgments, making it a potential proxy for evaluation.

**Audience:**

Yes

**Claims And Evidence:**

Yes

**Requested Changes:**

1. The justification of the paper contributions and focuses.
2. The justification of using CHIRP over Chatbot Arena.
3. The justification of using AI raters for frontier VLMs.

Minor issues:
The labels in Fig. 1 are unclear: 1. removing 10^0; 2. add labels to the right figure.

**Strengths And Weaknesses:**

The paper offers extensive analyses, with additional insights provided in the appendix. I also appreciate the authors' transparency regarding the limitations of CHIRP and their openness about Robin’s training data, hyperparameters, and evaluation examples.

Here are some detailed weaknesses:

**Scope and Focus**
- The paper covers too many topics, making it somewhat unfocused. It discusses:
1. A new family of VLM designs (Robin).
2. Scaling laws of VLMs.
3. Limitations of current VQA benchmarks.
4. A new benchmark (CHIRP).
5. AI vs. human evaluation alignment.

Each of these is a major topic within the VLM community. Addressing all of them in a single paper results in a lack of depth in any single area. A more structured focus, particularly on benchmarking, would improve clarity and impact.

- If CHIRP is the primary contribution based on my understanding, naming the paper after Robin seems misleading. A more suitable title should emphasize CHIRP as the benchmark, e.g. CHIRP: xxx instead of Robin: xxx.

**Limitations of Existing Benchmarks**
- Many of the identified flaws in current VQA benchmarks (e.g., incorrect ground-truth labels, limitations in evaluation strategies) are well-known in the VLM community. For example, the original TextVQA paper already highlighted: 1. Imperfect OCR systems. 2. Low agreement among annotators (only 22.8% of answers had perfect consistency). The existence of ambiguous questions (~5% of TextVQA lacks a clear correct answer, aligning with findings in this paper). While these insights help motivate the need for CHIRP, the paper dedicates too much space to restating known issues instead of offering deeper analysis or novel solutions.
- The discussion on prompting strategies as a source of evaluation inconsistency may be misframed. Differences in model behavior with various prompts are more about optimizing VLMs rather than flaws in benchmark design. Recent LLMs (e.g., OpenAI-O1, DeepSeek-R1) have already explored reinforcement learning and reasoning token strategies to improve performance. In datasets emphasizing commonsense reasoning (e.g., GQA), prompt-based long vs. short strategies might have limited impact, further complicating the interpretation of these variations as a benchmarking flaw.

**CHIRP Benchmark**

- CHIRP is a small-scale benchmark (102 questions), which limits its reliability. Other small-scale benchmarks, like LLaVA-Bench, were justified by the absence of alternative open-ended benchmarks at the time, and MM-Vet offers broader multi-capability VLM evaluations without relying on AI raters. In contrast, CHIRP does not significantly improve over existing benchmarks in scale or design.
- The pairwise ELO-based rating system is highly similar to Chatbot Arena (https://lmarena.ai/), a large-scale, community-driven benchmark where users compare LLM/VLM responses. Chatbot Arena has several advantages over CHIRP already: 1. It does not rely on AI raters. 2. It gathers large-scale user ratings from real-world interactions. 3. It allows open-ended user-generated inputs and questions.
Given these points, CHIRP appears to be a more restricted version of existing evaluation methods rather than a real improvement.
- The reliance on AI-based rating in CHIRP is problematic, as the paper itself finds that AI evaluations can diverge from human judgments. This raises concerns about its usefulness for future VLM evaluation. For example, if GPT-5V surpasses GPT-4V, using the latter as an evaluator would be invalid. A benchmark should ideally be independent of the models it evaluates, yet CHIRP ties its scoring to a potentially outdated AI judge.

---

> ### Author Response · Authors · 2025-03-18
> **Rebuttal**
>
> Dear Reviewer,
>
> Thank you for dedicating your time to thoroughly examine and assess our paper.
> Below we wish to address the points you made and how we are currently fixing them in the paper. We will submit a revision of the paper as soon as all the changes are made. If there is any point which we misunderstood please let us know.
>
> 1. We will refine the title, introduction and early sections to emphasize CHIRP as a central contribution, particularly its role in evaluating open-ended questions without ground-truth answers. This will ensure clearer positioning within the broader landscape of VLM benchmarks.
>
> 2. CHIRP is a benchmark, whereas Chatbot Arena is a ranking framework requiring authors to maintain an API access to their models. Chatbot Arena is not replicable, operates as a black box, and does not specifically focus on long-form responses. The two serve complementary roles: Chatbot Arena provides broad performance comparisons, while CHIRP is a benchmark specifically designed to evaluate open-ended questions, particularly in cases where there is no single ground truth answer.
>
>
> 3. We acknowledge that human evaluation remains essential for frontier VLMs. Using AIs to evaluate stronger AI models is an open research question, and we frame this as an area for future work.
> We will make this point explicitly clear in the paper.
> However, we'd like to emphasize that human evaluations remain a viable and practical metric — they are not prohibitively expensive when conducted on small scale evaluations.

---

### Review · Reviewer_d8EL · 2025-03-04

**Summary Of Contributions:**

This paper presents an empirical study on scaling laws in Vision-Language Models (VLMs), a critique of standard VQA benchmarks, and the introduction of CHIRP, a new long-form evaluation dataset.

1. Scaling Laws in VLMs (Robin Suite)
    1. The Robin suite, a collection of vision-language models of different VE & LLM combinations, was trained to explore the LLM and vision encoder scaling in multimodal models.
    2. Experiments show that LLM scaling improves performance, while vision encoder scaling shows inconsistent effects, confirming prior LLaVA-family research rather than introducing new insights.
2. Benchmark Limitations and Statistical Analysis
    1. It points out that standard VQA benchmarks (GQA, TextVQA) suffer from label noise, short-answer biases, and flawed scoring.
    2. The paper quantifies these issues using statistical methods (Elo ranking, agreement analysis, dataset noise evaluation).
3. CHIRP: A New Long-Form Evaluation Benchmark
    1. CHIRP introduces long-form Q&A, pairwise preference scoring (Elo ranking), and DALL·E-generated images for a more nuanced evaluation.
    2. CHIRP makes LLM scaling benefits more apparent, but vision encoder scaling remains weak.

**Audience:**

Yes

**Broader Impact Concerns:**

No ethical concerns were identified in aspects such as research methods, data usage, and model applications; the data sources are clear, and the research process adheres to ethical norms without discriminatory impacts or fairness - related issues.

**Claims And Evidence:**

Yes

**Requested Changes:**

1. Clarify the main focus and improve readability by restructuring the narrative.
2. Compare findings with well-recognized model families (e.g., LLaVA, Qwen-VL, LLaMA) to improve generalizability.
3. Strengthen justification for CHIRP, explaining why a new dataset is necessary instead of improving existing benchmarks.
4. Use stronger AI judges (e.g., GPT-4o, LLaVA-OneVision, LLaVA-Critic instead of GPT-4V, LLaVA-34B) and discuss evaluation limitations.
5. Provide practical takeaways for long vs. short answer evaluation beyond confirming known effects.
6. Analyze CHIRP’s scalability issues, and potentially propose some solutions.

**Strengths And Weaknesses:**

## strength
**Thorough Benchmark Evaluation and Statistical Analysis:** The paper identifies limitations in standard VQA benchmarks (label noise, ambiguous short answers, string-matching biases) and quantifies them using statistical methods. It employs Elo ranking for pairwise model comparison, agreement/contradiction analysis for AI judge reliability, and noise analysis for annotation inconsistencies. These techniques enhance the credibility of the benchmark critiques and ensure a data-driven, statistically robust evaluation framework.

**Introduction of CHIRP, a high quality Long-Form Evaluation:** CHIRP provides an open-ended evaluation dataset that captures reasoning depth beyond standard VQA benchmarks.

- It uses pairwise preference scoring (Elo ranking) instead of rigid accuracy-based metrics, reducing reliance on rigid accuracy-based metrics.
- It uses DALL·E-generated images that can ensure controllable distributions, reduce dataset biases and prevent benchmark leakage for more reliable evaluation.
- The evaluation instruction is carefully crafted and aligns with some popular RL evaluation strategies. This consistency with existing evaluation strategies adds credibility to the prompt design.

## Weakness
**Limitations of AI-Centric Evaluation and Reliance on non-scalable Human Annotation:** The paper employs these models as “critics,” but their limited understanding can lead to biased or incorrect verdicts—especially if they cannot solve a question or if the VLM under evaluation is *stronger*. Moreover, the paper uses older or non-specialized models (GPT‑4V, LLaVA‑34B) rather than the strongest current options (e.g., GPT‑4o, LLaVA‑OneVision, LLaVA‑Critic). While the authors measure judge reliability (tracking preference contradictions) and fall back on non-scalable human comparisons in key matchups, this does not fully resolve concerns about AI-centric scoring. Human annotation could similarly improve standard benchmarks (GQA, TextVQA), undermining the necessity of a new dataset, and leaving open how future, more advanced VLMs should be fairly evaluated by weaker AI critics.

**Robin’s scaling law findings are not new:** The scaling trends observed in this paper have already been reported in prior LLaVA studies, as Robin follows a similar architecture. For example, LLaVA-Next (May 2024) shows that VLM performance is primarily driven by LLM scale and capability [(source)](https://llava-vl.github.io/blog/2024-05-10-llava-next-stronger-llms/). Additionally, LLaVA-Next Ablation Studies (May 2024) confirm that LLM scaling has a greater impact than vision encoder scaling, with resolution and tokenization playing a more significant role than VE size [(source)](https://llava-vl.github.io/blog/2024-05-25-llava-next-ablations/). Given these prior findings, the Robin suite largely reconfirms known scaling trends rather than providing new insights, limiting its contribution.

**Low Readability and Lack of Focus:** The paper is difficult to read due to a lack of clear focus. It presents multiple overlapping themes—scaling laws, benchmark limitations, and the CHIRP dataset—without a strong central narrative. The transitions between these sections feel disjointed, making it unclear what the main takeaway is. Additionally, the presentation of findings is dense, with extensive statistical analysis that, while rigorous, detracts from clarity rather than reinforcing key insights.

---

> ### Author Response · Authors · 2025-03-18
> **Rebuttal**
>
> Dear Reviewer,
>
> We truly appreciate you taking the time to carefully review and provide feedback on our paper.
> Below we wish to address the points you made and how we are currently fixing them in the paper. We will submit a revision of the paper as soon as all the changes are made. If there is any point which we misunderstood please let us know.
>
> 1. To improve readability and strengthen the central narrative, we will restructure the paper to emphasize our key contribution: CHIRP’s ability to capture nuanced differences in long-form, open-ended responses that existing evaluation methods overlook.
> This will start with a review of the title, as suggested by reviewer 1XKB, to highlight CHIRP's importance and relevance.
> We will also highlight that our benchmark can detect subtle parameter differences between models, which motivates our testing on a VLM scaling suite.
>
>
> 2. We acknowledge the importance of comparing our findings with well-recognized model families such as LLaVA, Qwen-VL, and LLaMA to improve generalizability. However, these model families do not provide a controlled scaling study, as the differences between their released versions often involve multiple architectural or training modifications beyond just scaling. In some cases, only two model variants are available with a single parameter change, making it difficult to isolate the effects of scaling alone.
> In contrast, our work introduces a scaling suite that systematically evaluates VLMs across multiple scales while controlling for confounding factors. This approach allows us to make more precise observations about the impact of scaling on model performance
>
>
> 3. While there exists benchmarks like LLaVA-Bench which elicit open-ended responses, they often lack thought-provoking and diverse questions that challenge a model’s perception and contextual understanding. For example, most of the open-ended questions in LLaVA-Bench are of the form: "describe the image...".
> CHIRP addresses this gap by generating images specifically designed to elicit complex, open-ended responses across a wide range of scenarios.
> We can explicitly compare CHIRP with existing benchmarks, highlighting its unique contributions in question design and diversity. Let us know if we should include this kind of comparison in the paper if you believe it will clarify why CHIRP is necessary as a complementary resource rather than an extension of prior benchmarks.
>
>
> 4. We acknowledge the importance of using more advanced AI judges (e.g., GPT-4o (June 2024), LLaVA-OneVision (August 2024), LLaVA-Critic (Octobre 2024)). These were not originally used as the work was already well underway when they were released.
> However, our results demonstrate that even with less powerful models, there is a strong correlation between AI-based evaluations and human judgments, supporting the validity of using these evaluators as proxies. Nonetheless, if you still believe evaluating on more recent AI judges is necessary, we are open to doing so.
>
>
> 5. While the differences between long and short-form evaluation are known, they have not been systematically quantified, particularly in vision-language tasks. Our study provides empirical evidence, aligning with very recent findings in LLMs (e.g., DeepSeek R1). We will clarify this takeaway and may move details to the appendix for better readability.
> Ultimately the goal in including these experiments was to further motivate the need for evaluating on long-form open-ended responses.
>
> 6. Our goal was to demonstrate that AI-based evaluation can serve as a reliable proxy for human evaluation, thereby limiting scalability concerns. We will clarify that our study does not introduce new scalability issues but rather aims to address them. Furthermore, we are also working on scaling the CHIRP benchmark following the detailed methodology written in the paper.

---

### Review · Reviewer_pmN3 · 2025-03-08

**Summary Of Contributions:**

This paper first constructs a suite of experiments, which includes vision encoders and language models of various sizes. The authors do not observe a scaling law across model sizes when evaluated using existing benchmarks. They then take a step further and identify intrinsic flaws in these benchmarks and evaluation methods, such as incorrect ground truth answers. To address these issues, the authors create a new dataset called CHIRP, which consists of 104 open-ended questions and AI-based evaluations. When evaluated using CHIRP, the authors observe a scaling law across model sizes, and the proposed AI-based evaluation shows consistency with human evaluations.

**Audience:**

Yes

**Claims And Evidence:**

Yes

**Requested Changes:**

See weakness

**Strengths And Weaknesses:**

### Strengths
1. This paper is well-written and easy to read. The exploration process and the design of experiments are systematic.
2. The investigated problem is interesting and highly valuable.
3. The proposed points in this paper are exhaustively validated by meticulously designed experiments.

### Weaknesses
1. The datasets used in pre-training, such as CS-558K, and in visual instruction tuning, such as LLaVA-665K, are quite small. Based on my experience, this may have a significant impact on the experimental results and the conclusions drawn from them.
2. Following the concern above, scaling only the model size without increasing the size of the training datasets may also contribute to the absence of a scaling law. I would like to see the authors' comments on this.
3. During the evaluation, the authors chose traditional benchmarks like TextVQA and VQAv2. The drawbacks of these datasets and evaluation methods have already been identified by previous works. I am curious why the authors did not choose more recent benchmarks, such as MMBench[1] and MathVista[2].
4. In the first experiment to investigate the scaling law, I hope the authors can provide results on more diverse datasets, such as MMBench, OCRBench[3], and MathVista.
5. AI-based evaluation is not new and should not be treated as a contribution of this paper.
6. The size of the dataset proposed by this paper is quite small

[1] MMBench: Is Your Multi-modal Model an All-around Player?

[2] MathVista: Evaluating Mathematical Reasoning of Foundation Models in Visual Contexts

[3] OCRBench: On the Hidden Mystery of OCR in Large Multimodal Models

---

> ### Author Response · Authors · 2025-03-18
> **Rebuttal**
>
> Dear Reviewer,
>
> Thank you very much for taking the time to thoroughly go through and review our paper.
> Below we wish to address the points you made and how we are currently fixing them in the paper. We will submit a revision of the paper as soon as all the changes are made. If there is any point which we misunderstood please let us know.
>
> 1. While our datasets are relatively small, they align with prior work such as LLaVA, which used the same data for a SoTA 13B model.
> Given that our models are smaller in scale, we believe the dataset sizes we used are appropriate for the purpose of conducting meaningful experiments.
> That being said, we acknowledge this limitation.
>
> 2. We agree that increasing model size without proportionally scaling training data can impact scaling laws.
> However, our approach follows established practices in scaling experiments, such Kaplan et al.'s work on scaling laws, which suggest that only one variable should be changed at a time to isolate its effect. In our case, we focused on scaling the model's size while keeping the dataset constant to specifically examine its impact.
> It is our belief that if the data is sufficient to train the LLaVA 1.5 13B model, then it is sufficient to train any smaller models, as we do here, without being a limiting factor in their performance.
>
> 3. We initially based our selection on widely used benchmarks (e.g., MM-Vet, LLaVA-Bench) but acknowledge the importance of newer datasets. We attempted to run MMBench but faced server issues on their side and will retry. We will also incorporate MathVista for a broader evaluation.
>
> 4. We appreciate the suggestion and have begun evaluating the models to include additional benchmarks such as OCRBench and MathVista to strengthen our analysis.
>
> 5. We did not intend to present AI-based evaluation as a novel contribution of our work, and we apologize for any such unintended implications.
> We will read through the paper to remove any places where we make such an implication.
> We acknowledge that prior benchmarks such as MM-Vet and LLaVA-Bench have already employed similar methodologies. Instead, our focus is on introducing a new AI-based evaluation method specifically designed to address identified shortcomings in existing approaches.
> Furthermore, our work provides a thorough comparison between AI and human evaluations on our benchmark, offering insights into their alignment and potential limitations.
> Our primary objective in exploring AI-based evaluation is to assess its applicability on our benchmark rather than to position it as a standalone contribution.
>
> 6. With regard to the size of the benchmark, we believe a smaller dataset is acceptable for CHIRP as it prioritizes quality over quantity. Each question is carefully curated, allowing for more complex, open-ended evaluations that better assess advanced vision-language capabilities. This approach, similar to respected benchmarks like LLaVA-Bench and MM-Vet, provides more insightful evaluation than larger datasets with simpler, automated assessments. Additionally, the smaller size makes vision-language model-based evaluation more practical and cost-effective.
> However, we understand the concerns regarding scale and are actively working on an extended version with larger coverage.

---

### Author Response · Authors · 2025-03-25
**PDF Revision**

Dear Reviewers,

Thank you for your patience, we have taken all of your feedback into account. In particular, we have addressed the main concern about the confusing narrative, and have significantly restructured the paper to clarify our main contribution.

We believe that this version comprehensively addresses your concerns but please let us know if you find any issues. We very much appreciate your feedback.

---

### Decision · Action_Editor_tyTA · 2025-04-13

**Recommendation:** Reject

**Comment:**

Initially, the paper, “Robin: a Suite of Multi-Scale Vision-Language Models and the CHIRP Evaluation Benchmark,” introduces Robin, a suite of VLMs (LLaVA-like models). Via Robin, the authors identified a series of issues existing in VQA benchmarks, which motivates the authors to propose CHIRP, a new benchmark with 102 questions.

Given the original draft, all reviewers acknowledged systematic and thorough evaluations conducted by the authors. However, all reviewers expressed several concerns, as summarized below:

* The main focus of the original draft is unclear. Should it be Robin or CHIRP or experimental discoveries?

* Several findings, such as the limitations of existing VQA benchmarks and the scaling behavior of Robin, appear to be duplicative of well-known observations in the VLM community.

* The proposed CHIRP benchmark raises concerns due to its limited scale, dependence on AI-generated ratings, and lack of clear advantages over existing benchmarks.

After the rebuttal, the authors updated and restructured the paper with a new focus, “CHIRP: A Fine-Grained Benchmark for Open-Ended Response Evaluation in Vision-Language Models,” which emphasizes on their main contribution (i.e., the benchmark).

As a result, Reviewer 1XKB voted for leaning accept, while the other two Reviewers voted for leaning reject. The reviewers are mainly concerned about (1) the experimental setup needs further refinement (e.g., adding more modern VLM results on CHIRP, support of leaderboard evaluation), (2) the limitation of small-scale experiments, and (3) lack of narrative focus (the restructured draft still lacks a clear takeaway).

Given the reviewers’ unaddressed concerns, the Action Editor could not recommend acceptance of the paper. That said, the Action Editor concurs with the reviewers that the proposed CHIRP benchmark shows potential as a useful evaluation tool. However, the current draft falls short in demonstrating its generality, readiness for adoption, and necessity. The authors are encouraged to revise the paper and refine the benchmark by incorporating the reviewers' suggestions.

**Audience:**

The proposed benchmark, CHIRP, has the potential as a useful evaluation tool for VLMs, and is likely to be of interest to the TMLR's audience.

**Claims And Evidence:**

In the beginning, the submission covered multiple topics, making it hard to understand the main contribution. After the rebuttal, the authors significantly revised the draft to focus more on the proposed evaluation benchmark, CHIRP. However, current version still falls short in demonstrating CHIRP's generality (e.g., additional results from various VLMs are needed, instead of only Robin), and its readiness for adoption (e.g., small-scale limitations and insufficient guidance for external researchers to adopt this benchmark). Furthermore, reviewers continue to raise concerns about the clarity of the paper, noting that the main focus is still unclear and that Robin scaling findings offer limited novelty, as similar observations have been reported in prior work.

**Resubmission Of Major Revision:**

The authors may consider submitting a major revision at a later time.

---

> ### Author Response · Authors · 2025-05-05
>
> Dear Reviewers,
>
> Thank you very much for taking the time to review the updated revision of our paper.
>
> From the current feedback, it is our understanding that we should:
> - increase the size of CHIRP to 500
> - run CHIRP between modern models (GPT-4o, InternVL, Gemini)
> - make a leaderboard with these models
> - continue to clarify the writing of the paper
>
> Would these changes be sufficient?